# None to Optima in Few Shots: Bayesian Optimization with MDP Priors

## Abstract

Bayesian Optimization (BO) is an efficient tool for optimizing black-box functions, but its theoretical guarantees typically hold in the asymptotic regime. In many critical real-world applications such as drug discovery or materials design, where each evaluation can be very costly and time-consuming, BO becomes impractical for many evaluations. In this paper, we introduce the Procedure-inFormed BO (ProfBO) algorithm, which solves black-box optimization with remarkably few function evaluations. At the heart of our algorithmic design are Markov Decision Process (MDP) priors that model optimization trajectories from related source tasks, thereby capturing procedural knowledge on efficient optimization. We embed these MDP priors into a prior-fitted neural network and employ model-agnostic meta-learning for fast adaptation to new target tasks. Experiments on real-world Covid and Cancer benchmarks and hyperparameter tuning tasks demonstrate that ProfBO consistently outperforms state-of-the-art methods by achieving high-quality solutions with significantly fewer evaluations, making it ready for practical deployment.

## 1 Introduction

Bayesian Optimization (BO) is an efficient machine learning–based approach for solving global optimization problems of black-box functions where the objective functions can be highly non-convex, and the functional form or derivative is not necessarily available. Owing to its ability to optimize black-box functions, BO is particularly suited for and has been applied in many critical real-world applications such as hyperparameter optimization (Snoek et al., 2012; Wu et al., 2020), neural architecture search (Kandasamy et al., 2018; Zhou et al., 2019), drug discovery (Pyzer-Knapp, 2018; Shields et al., 2021), and materials design (Khatamsaz et al., 2023; Tian et al., 2025).

In BO settings, the objective function can only be accessed by sequential, expensive and time-consuming evaluations. At each iteration all previous evaluations guide the selection process for the next query of the objective function. To study the data efficiency of BO, numerous theoretical studies focused on establishing asymptotic regret bounds (Srinivas et al., 2012; Chowdhury & Gopalan, 2017), such as cumulative regrets and simple regrets. However, asymptotic efficiency often proves impractical in real-world experimental design scenarios, primarily due to the high cost and long turnarounds of each function evaluation. In penicillin manufacturing, for example, the pharmaceutical goal is to identify the optimal experimental control parameters (e.g., temperature, humidity, biomass concentration) that maximize yield. Unfortunately, each evaluation via the wet labs can take days or weeks and incur substantial costs, severely limiting the number of feasible evaluations per year. Notably, Liang & Lai (2021) reported that even state-of-the-art BO algorithms require around 1,000 iterations to converge in a penicillin production simulator, equivalent to nearly 20 years if each evaluation takes one week. Also, Aldewachi et al. (2021) reported that developing a new drug cost approximately $1.3 billion USD on average in 2018, while a failed Alzheimer's disease drug development program could take up to 5 years and cost as much as $2.5 billion.

Therefore, to significantly accelerate the scientific discovery processes mentioned above, an urgent and important question is:

> Can we design a BO algorithm that is able to find the global optimum within very
> few shots, e.g., fewer than 90 evaluations?

In this paper, we answer this question affirmatively by proposing a few-shot BO method, the **Pro**cedure-in**f**ormed **B**ayesian **O**ptimization (PROFBO). How does PROFBO address the problem above that initially seems impossible? The key insight behind PROFBO is that, although the target task permits only a few evaluations, related source tasks often have existing evaluations that can serve as rich sources of prior information. Leveraging these existing evaluations enables us to both design and accelerate a BO algorithm for the target task. In practice, these related source tasks can be the docking scores of a set of molecules evaluated on different receptors (Liu et al., 2023), or evaluations of the same supervised-learning loss function on different datasets (Pineda-Arango et al., 2021).

The most straightforward way to exploit such source information is to directly predict the shape of the target objective function, as studied in Wistuba & Grabocka (2021); Falkner et al. (2018); Wang et al. (2024). In this paper, however, we propose a framework that optimizes the target objective function by

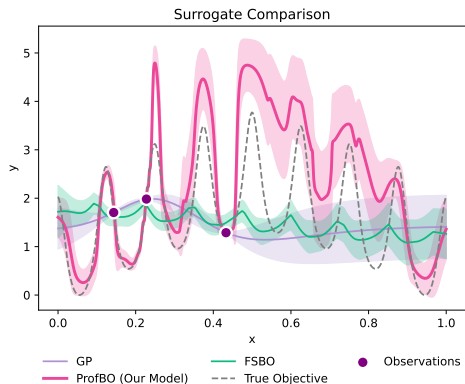

Figure 1: Comparison of function predictions with only 3 observation points. PROFBO models the true objective function curve significantly better than GP model and FSBO algorithm (Wistuba & Grabocka, 2021).

leveraging optimization trajectories of related source tasks in principled Bayesian perspective. Specifically, we use Markov Decision Process (MDP) (Bellman, 1958) priors to model source optimization trajectories, thereby capturing procedural knowledge on efficient optimization. We then embed these MDP priors into a Prior-Fitted Neural Network (PFN) (Müller et al., 2022) and employ Model-Agnostic Meta-Learning (MAML) (Finn et al., 2017) for fast adaptation to new target tasks. The whole PROFBO framework works efficiently even with very few evaluations. See Figure 1 for an example showing that with only 3 evaluations, our PROFBO framework models the true objective function significantly better than the standard Gaussian Process (GP) model and the FSBO algorithm (Wistuba & Grabocka, 2021) .

**Contributions.** Our contributions are summarized as follows:

1. With very few shots, our PROFBO framework can accurately and efficiently identify global optima of black-box functions. Its modular design allows easy adaptation to various configurations and input types by retraining the MDP priors part only. Both make PROFBO highly practical for a wide range of scientific and engineering applications.

2. While optimization trajectory information from source tasks is needed, the Bayesian framework underlying PROFBO *eliminates* all manual steps in prior design and posterior inference via the universal PFN inference.

3. The core design of the PROFBO algorithm lies in its use of MDP priors, which model optimization trajectories from source tasks, thereby capturing procedural knowledge of efficient optimization. These MDP priors are then embedded into a PFN, and MAML is employed for rapid adaptation to new target tasks.

4. We establish new real-world Covid and Cancer benchmarks for few-shot BO problems and show that PROFBO achieves better performances than all existing state-of-the-art baselines.

## 2 RELATED WORKS

**Few-Shot Bayesian Optimization.** In few-shot BO, people often leverage the knowledge in the evaluations of related tasks to improve the performance of BO. One common approach is meta-surrogate design, whose goal is to transfer-learn a BO surrogate that captures the shared features of multiple tasks' response surfaces. The meta-surrogates can be synthesized from surrogates of related tasks. Falkner et al. (2018); Schilling (2016); Wistuba et al. (2016) used an ensemble surrogate with weights based on metrics like task similarity or model uncertainty from related task evaluations. Meta-prior design also induces the corresponding meta-surrogate, where knowledge from related

tasks is transferred to the prior distribution of the target task. Swersky et al. (2013); Poloczek et al. (2016); Yogatama & Mann (2014); Law et al. (2019); Tighineanu et al. (2022); Wistuba & Grabocka (2021) proposed to design novel kernels for GP functional priors. Wang et al. (2024) proposed to obtain a target task prior by minimizing the KL divergence between it and its related tasks. Perrone et al. (2018) designed a linear function prior for the target objective whose feature is processed by a meta-trained neural network, and the inference is facilitated by Bayesian linear regression. However, most existing work relies on manually designed ensemble weights or tractable priors, and meta-learns only the response surface, while our approach *eliminates* those manual designs through the universal PFN inference, and introduces a trajectory surrogate that captures procedural information.

Acquisition function design is another important approach of few-shot BO, which also employs ensemble methods (Wistuba et al., 2017), similar to the surrogate design. Some work reframed BO as sequence modeling or Reinforcement Learning (RL). According to Bai et al. (2023), search space and initialization design (Wistuba et al., 2015a; Perrone et al., 2019; Li et al., 2022; Feurer et al., 2015; Wistuba et al., 2015b) also provide important information for few-shot BO, and their methods primarily leverage source-domain information to provide a warm start for the target task or narrow down the search space, which is far from the techniques used in our work.

**Meta-Optimization with Sequence Modeling.** An emerging trend in leveraging meta-data in optimization is to frame optimization as a generic sequential decision process, thus the knowledge transfer is not limited to BO components. Those methods have demonstrated state-of-the-art results in various hyperparameter optimization and drug discovery problems. The meta-acquisition learning method (Volpp et al., 2020; Hsieh et al., 2021) reframes BO (or optimization) as a MDP. Optimization trajectories are generated during their own training processes. They meta-train a RL agent on evaluations of related tasks and conduct optimization on target tasks. Iwata (2021); Maraval et al. (2023) trained the acquisition function and surrogate model end-to-end, defining the state space solely as historical evaluations, which is consistent with this work. Unlike these approaches, we employ a lightweight RL agent to extract procedural experience, and generate optimization trajectories separately from the training process. And our agent is not used to optimize the target tasks, but serves as the prior distribution of the optimization trajectory.

The Transformer (Vaswani et al., 2017) architecture excels in sequence prediction and in-context learning. The Transformer Neural Process (Nguyen & Grover, 2022)-based models have shown strong performance to conduct meta-learning for complicated priors, demonstrating high potential for many BO tasks. Prior-fitted Neural Networks (PFNs) excel among these methods, demonstrating both the flexibility to integrate user priors (Müller et al., 2023) and the ability to efficiently perform in-context freeze–thaw BO (Rakotoarison et al., 2024). Ramos et al. (2023); Liu et al. (2024) leveraged the knowledge in Large Language Models (LLMs) to conduct in-context learning. Chen et al. (2022); Nguyen et al. (2024) framed optimization as a sequence prediction task, using LLMs to jointly predict the next query and response, leveraging evaluations of other optimization algorithms. Our approach also uses Transformer to facilitate in-context learning, but adopts a modular design that allows easy adaptation by retraining only the MDP priors for various configurations and input types.

## 3 PRELIMINARIES

### 3.1 PROBLEM STATEMENT

Let $[n]$ denote the set $\{1, \cdots, n\}, n \in \mathbb{N}^+$. We consider the target problem of finding the global optimum of a black-box function $f : \mathcal{X} \to \mathbb{R}$:

$$x^* = \arg \max_{x \in \mathcal{X}} f(x), \tag{1}$$

where $\mathcal{X} \subseteq \mathbb{R}^d$ is the function domain and $d$ is the input dimension. $f$ is said to be a black-box function because its closed-form expression or the derivative is not necessarily known, and it can be a non-linear non-convex function. We learn from $f$ only through sequential noisy evaluations. Throughout $T$ iterations, at each iteration $t \in [T]$, its evaluation is given by $y_t = f(x_t) + \zeta_t$ where $\zeta_t$ is the noise. Our goal is to solve this problem within a few shots, e.g., $T \leq 20$.

Given the limited information gained from each evaluation, we assume access to historical source knowledge that can accelerate the target optimization process, enabling completion within only a

few evaluations. Formally, historical knowledge can be $D^{(i)} = \{x_\tau^{(i)}, y_\tau^{(i)}\}_{\tau=1}^{n_i}, \forall i \in [N]$ generated by $N$ source black-box functions $f^{(1)}, \cdots, f^{(N)} : \mathcal{X} \to \mathbb{R}$ where $n_i$ denotes the evaluation length of $f^{(i)}$. Additionally, we define $p(\mathcal{T}^{(i)})$ as the process of sampling from $f^{(i)}$, which can also be comprehended as the distribution of optimization trajectories of $f^{(i)}$ under some policy. $p(\cdot)$ or $p(\cdot|\cdot)$ denotes a (conditional) probability density function or its corresponding distribution. If $p$ is a distribution of a function $f : \mathcal{X} \to \mathbb{R}$, it will specify the distribution of $f(x)$ with probability density function $p(\cdot|x)$ where $x \in \mathcal{X}$.

## 3.2 BACKGROUND

**Bayesian Optimization (BO).** To solve eqn. (1), BO usually assumes $f$ is drawn from a functional prior, e.g., a GP prior. At iteration $t$, conditioning on historical evaluations $D_{t-1} = \{x_\tau, y_\tau\}_{\tau=1}^{t-1}$, we denote the posterior predictive distribution as $p(\cdot|x, D_{t-1})$. Then the next query $x_t$ is chosen by $x_t = \arg\max_{x \in \mathcal{X}} \alpha_t(x)$ where $\alpha_t$ is the acquisition function built on $p(\cdot|x, D_{t-1})$. Upper Confidence Bound (UCB) (Srinivas et al., 2012), Expected Improvement (EI) (Jones et al., 1998), Thompson sampling (Thompson, 1933), and knowledge gradient (Frazier et al., 2008) are commonly used as acquisition functions in practice.

**Prior-Fitted Neural Networks (PFNs).** The motivation of applying PFNs (Müller et al., 2022) in BO is to leverage the meta-trained Transformer architecture to perform principled Bayesian inference in a single forward pass, enabling superior few-shot learning and accurate predictions with interpretable uncertainty quantification. It outperforms traditional GP (see Figure 1), offering greater flexibility in the choice of prior and faster inference by avoiding the matrix inversion in GP posterior inference, thereby enabling our proposed MDP priors (Section 4.2). Given any observation set $D$ and an inference location $x \in \mathcal{X}$, the PFN $q_\theta(\cdot|x, D)$ approximates $p(\cdot|x, D)$ through a bar distribution. In practice, we evaluate a sequence of queries $\{x_i\}_{i=1}^m$ in parallel. Figure 2 illustrates the forward pass and attention mask with three contextual observations and two inference locations. Our settings of PFN's regression head and network structure are shown in Appendix A.1.

**Markov Decision Process (MDP).** An MDP (Sutton & Barto, 2018) is typically defined as a 5-tuple $\mathcal{M} = \{\mathcal{S}, \mathcal{A}, \mathcal{P}, \mathcal{R}, \gamma\}$ where $\mathcal{S}$ is the state space, $\mathcal{A}$ is the action space, $\mathcal{P}$ is the transition probability space and $p(s_{t+1}|s_t, a_t) \in \mathcal{P}$ is the transition probability from state $s_t \in \mathcal{S}$ to state $s_{t+1} \in \mathcal{S}$ if the agent takes action $a_t \in \mathcal{A}$ at time $t$, $\mathcal{R} \subseteq \mathbb{R}$ is the reward space and $r_t(s_t, a_t) \in \mathcal{R}$ is the reward of taking action $a_t$ at state $s_t$, and $\gamma \in (0, 1]$ is a discount factor. A classical RL problem is to train an agent by finding the optimal policy $\pi^*$ that maximizes the expected cumulative discounted reward $\pi^* = \arg\max_\pi \mathbb{E}\left[\sum_{t=1}^{\infty} \gamma^{t-1} r_t\right]$ where the expectation is taken w.r.t. MDP.

**Model-Agnostic Meta-Learning (MAML).** MAML (Finn et al., 2017) is a gradient-based meta-learning framework designed to train a model $M_\theta$ that can rapidly adapt across a collection of tasks $\{\mathcal{T}^{(i)}\}_{i=1}^N$. Each task $\mathcal{T}^{(i)}$ specifies a loss $L^{(i)}(\theta)$. The meta-objective minimizes the meta loss defined as $L_{\text{meta}}(\theta) := \sum_{i=1}^N L^{(i)}(\theta - \beta \cdot \partial_\theta L^{(i)}(\theta))$ where $\beta$ is an inner step size. By descending with respect to the meta loss, the model learns the common internal representation across $N$ tasks.

## 4 THE PROFBO ALGORITHM

In this section, we show details of our PROFBO algorithm, a Bayesian framework that can find high-quality solutions to the black-box function optimization problem within very few shots where $T$ can be fewer than 90. The key design of PROFBO lies in how it can efficiently transfer the knowledge from historical optimization trajectories obtained from related source tasks to accelerate its optimization process in target task. The core procedure is summarized in Figure 2.

The PROFBO framework is grounded in a principled Bayesian perspective on optimization trajectories. Under the GP assumption in standard BO, at iteration $t$, all queries in $D_{t-1}$ are treated as being exchangable, ignoring the fact that $D_{t-1}$ is temporal-correlated, since it was generated from a BO procedure. To fully make use of this information to accelerate the BO process in the target task, we aim to construct a new surrogate model that replaces GP with a set of optimization trajectory priors, $p(\mathcal{T}^{(i)}), \forall i \in [N]$ (Figure 2 right, Section 4.2). To facilitate posterior inference under such a prior, we introduce the PFN model $q_\theta(\cdot|x, D_{t-1})$ as a proxy for $p(\cdot|x, D_{t-1})$ (Figure 2 left-top, Section 4.1). By leveraging observations from the source tasks $\{D^{(i)}\}_{i=1}^N$, MAML enables the PFN model

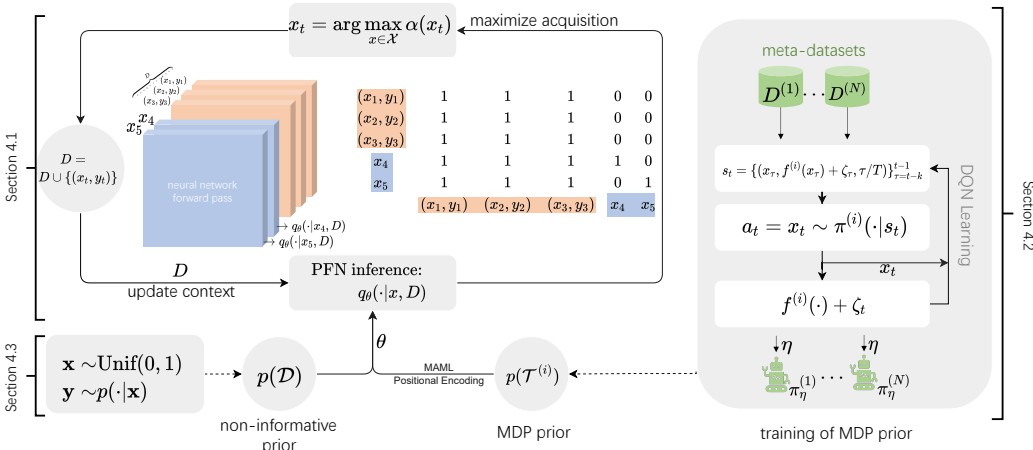

Figure 2: Overview of our PROFBO framework. [Left] The BO loop: a PFN (Müller et al., 2022) model pre-trained with a non-informative prior (e.g., GP) and fine-tuned with MDP priors from source tasks. Fine-tuning uses positional encoding and MAML (Finn et al., 2017) for better knowledge transfer. The PFN performs posterior inference on context $D$ via Transformer attention, with outputs interpreted as logits of a bar distribution. [Right] MDP prior training: for each meta-dataset $D^{(i)}$, a Deep Q-Network (DQN) (Mnih et al., 2013) policy generates optimization trajectories of the corresponding objectives. Right flowchart is attributed to Figure 1 in Volpp et al. (2020).

to rapidly adapt to the unknown trajectory of the target task $f$, while preventing it from learning spurious temporal correlations (Figure 2 left-bottom, Section 4.3).

## 4.1 THE PFN FRAMEWORK

First, we introduce the PFN framework used in the BO loop where the surrogate model $p(\cdot|x, D_{t-1})$ is obtained by updating the observation set $D_{t-1}$ and conducting posterior inference. PFN conducts simulation-based inference through the forward pass of a Transformer model. It directly takes $D_{t-1}, x$ as inputs and outputs a discrete approximation of $p(\cdot|x, D_{t-1})$. Each query-evaluation pair in $D_{t-1}$ attends to each other through the Transformer attention, and are attended by the position of the query $x$. The final layer of the query network is converted to logits of $p(\cdot|x, D_{t-1})$, through which we can compute approximated acquisition functions, such as UCB and EI.

We assume that the evaluations of $f$ is generated by the process $p(\mathcal{D})$, which is similar to $p(\mathcal{T}^{(i)})$ defined in Section 3 corresponding to a certain process sampling from $f$. To train the PFN to conduct posterior inference for $p(\mathcal{D})$, one only need to minimize the KL divergence between the PFN approximation $q_\theta(\cdot|x, D)$ and the ground-truth $p(\cdot|x, D)$, where $\theta$ is the model parameter and $D$ is any context. It is equivalent to a more tractable negative log-likelihood term $\mathbb{E}_{D\cup\{(x,y)\}\sim p(\mathcal{D})}[-\log q_\theta(y|x, D)]$ plus a constant, where $\{(x, y)\} \cup D$ are sampled from $p(\mathcal{D})$ [1]. In each gradient descent step, we aim to teach the model to make inference for context $D$ at $m$ inference locations, thus the step-wise loss is

$$\hat{\ell}_\theta(D \cup \{(x_i, y_i)\}_{i=1}^m) = \sum_{i=1}^m -\log q_\theta(y_i|x_i, D), \quad D \cup \{(x_i, y_i)\}_{i=1}^m \sim p(\mathcal{D}) \quad (2)$$

We use a fixed total sequence length for each training step, and $m$ in each step is chosen to randomly split the sequence. The flexible choice of $p(\mathcal{D})$ provides the possibility to infer any complicated while easy-to-generate prior. Thus, we construct simulators of $p(\mathcal{T}^{(i)})$ based on meta-datasets of related tasks $D^{(i)}, \forall i \in [N]$ (Section 4.2) and train the PFN with the supervised loss in eqn. (2), yielding a desired trajectory surrogate for BO.

---

[1]The derivation can also be found in Appendix A in Müller et al. (2022).

## 4.2 Modeling Optimization Trajectory with MDP

We propose a novel prior for Bayesian optimization that models $p(\mathcal{T}^{(i)})$, the prior distribution of optimization trajectories, with MDP (Bellman, 1958; Sutton & Barto, 2018). In this MDP, at time $t$, the action $a_t$ is defined as the next query point $x_t \in \mathcal{X}$. The state $s_t$ consists of the previous $k$ evaluations $\{x_\tau, f^{(i)}(x_\tau) + \zeta_\tau, \tau/T\}_{\tau=t-k}^{t-1}$, and the transition is adding a new evaluation $(x_t, f^{(i)}(x_t) + \zeta_t, t/T)$ while removing the oldest evaluation obtained at time $\tau = t - k$. The reward is the negative simple regret $r_t = \max_{\tau \leq t} f^{(i)}(x_\tau) - \max_{x \in \mathcal{X}} f^{(i)}(x)$. The agent thus performs $T$-shot optimization of $f$. We train an agent for each source task $f^{(i)}$ via a DQN policy (Mnih et al., 2013), and denote the resulting policy parametrized by $\eta$ as $\pi_\eta^{(i)}$ (Figure 2 right). Trajectories sampled according to policy $\pi_\eta^{(i)}$ and state transition defined above are then used to train PFNs.

The DQN agent approximates the optimal Q-function with $Q_\eta(s, a)$ and updates $\eta$ iteratively during training on dataset $D^{(i)}$ until surpassing random search (details in Appendix A.2). Its parametric nature allows efficient batched trajectory generation on GPUs, which is crucial for PFN training.

## 4.3 Model-Agnostic Meta-Learning for Trajectory Surrogate

Next, we show how PROFBO trains the PFN framework using MAML for trajectory surrogate, with details shown in Algorithm 1. Let $\text{GD}_\theta[f] := \theta - \beta \cdot \partial_\theta f$ denote the gradient descent operator where $\beta$ is a learning rate. In the pre-training stage (Line 1–4), we train the PFN conventionally using synthetic samples from common function priors $p(\mathcal{D})$, such as GPs. As shown in Müller et al. (2023), PFNs trained with non-trajectory functional priors are well-suitable BO surrogates, as they generate sensible mean predictions with uncertainty quantification. Therefore, $p(\mathcal{D})$ provides a warm start for the following stage. Moreover, since trajectories may not cover the entire response surface of the objective, training with $p(\mathcal{D})$ stabilizes regions unexplored by $p(\mathcal{T}^{(i)})$s. Each iteration involves gradient descent (Line 3) on the loss from a batch of samples (Line 2), enabling the PFN to infer across more diverse contexts.

---

**Algorithm 1** PFN training of PROFBO

---

**Inputs:** Prior over general distribution $p(\mathcal{D})$; prior over source tasks $\{p(\mathcal{T}^{(i)})\}_{i=1}^N$, numbers of iterations $K_1, K_2$, initialized PFNs parameter $\theta$.

1: **for** $j \in [K_1]$ **do**
2:      Sample $D \cup \{(x_i, y_i)\}_{i=1}^m \sim p(\mathcal{D})$
3:      Update $\theta = \text{GD}_\theta[\hat{\ell}_\theta(D \cup \{(x_i, y_i)\}_{i=1}^m)]$            ▷ standard gradient descent
4: **end for**
5: **for** $j \in [K_2]$ **do**
6:      sample a batch of priors $\Pi \subset \{p(\mathcal{T}^{(i)})\}_{i=1}^N$            ▷ sample from MDP prior
7:      **for** $p \in \mathcal{P}$ **do**
8:          sample $D \cup \{(x_i, y_i)\}_{i=1}^m \sim p$
9:          Compute $\theta_\pi = \text{GD}_\theta[\hat{\ell}_\theta(D \cup \{(x_i, y_i)\}_{i=1}^m)]$            ▷ with positional encodings
10:      **end for**
11:      Update $\theta = \text{GD}_\theta\left[\sum_{\pi \in \Pi} \hat{\ell}_{\theta_\pi}(D \cup \{(x_i, y_i)\}_{i=1}^m)\right]$            ▷ MAML update
12: **end for**

**Output:** $q_\theta$

---

In the fine-tuning stage (Line 5–12), we fine-tune the PFN using samples from all $p(\mathcal{T}^{(i)})$, $\forall i \in [N]$, incorporating both MAML (Finn et al., 2017) and positional encoding. We introduce the positional encoding to better capture sequential information in the MDP prior, unlike the original PFN (Müller et al., 2022) that omitted it for permutation-invariant priors. However, this also increases the risk of overfitting to spurious correlations. To address this, we combine positional encoding with MAML, which helps the model generalize well by focusing on common optimization patterns rather than overfitting task-specific details. Unlike its original purpose of warm-starting parameters, MAML in our method is creatively repurposed to extract features shared across trajectories. Fine-tuning with MAML involves three steps: sampling tasks (Line 8), computing task-specific updates (Line 9), and

optimizing the total adapted loss (Line 11). In Section 5.4, we show through an ablation study that both MAML and positional encoding are crucial for building a robust trajectory surrogate.

The training cost in the first stage is incurred only once, since the base PFN can be stored and later fine-tuned for future problems. Thus, the cost for future problems is reduced to the fine-tuning stage.

# 5 EXPERIMENTS

## 5.1 EXPERIMENTAL SETUP

**Baselines.** We compare PROFBO with several few-shot BO methods using different techniques: BO with meta-learned GP (META-GP), few-shot deep kernel surrogate learning (FSBO) (Wistuba & Grabocka, 2021), pure transformer neural processes (TNP) (Nguyen & Grover, 2022; Müller et al., 2022), end-to-end meta-BO with transformer neural processes (NAP) (Maraval et al., 2023), random search (RANDOM), and BO with standard GP (GP). Table 1 shows a technical summary of them except RANDOM and GP since they do not use any listed techniques.

In TNP, we meta-train a TNP [2] on original meta-data instead of MDP priors; for comparison, we also test a variant with MAML and positional encoding, denoted as TNP+. Similarly, we pre-train META-GP kernel parameters with meta-data before testing, and introduce a meta-trained version of our MDP prior, denoted as Meta-Acquisition Function (MAF), to align with RL-based methods. We also include OPTFORMER (Chen et al., 2022) when possible [3]. As summarized in Table 1, these baselines fall into two main categories: *meta-surrogate learning* (META-GP, FSBO, TNP), which learn BO surrogates without sequential information, and *meta-trajectory learning* (MAF, OPTFORMER), which directly model trajectories. PROFBO and NAP use both approaches.

| Techniques | META-GP | FSBO | TNP | MAF | NAP | OPTFORMER | PROFBO (Ours) |
|---|---|---|---|---|---|---|---|
| Meta-learn traj. | ✗ | ✗ | ✗ | ✓ | ✓ | ✓ | ✓ |
| Meta-learn surrog. | ✓ | ✓ | ✓ | ✗ | ✓ | ✓ | ✓ |
| MAML | ✗ | ✗ | ✗ | ✗ | ✗ | ✗ | ✓ |
| Positional encoding | ✗ | ✗ | ✗ | ✗ | ✗ | ✓ | ✓ |

Table 1: Technical summary of all compared algorithms except RANDOM and GP. "✓" and "✗" denote a certain technique is used in an algorithm or not.

**Evaluations.** The objective function range of each task is normalized to $[0, 1]$, so the regret at iteration $t$ is defined as $1 - \max_{0 \leq \tau \leq t} f(x_\tau)$ and we use the log-scaled version in our results. The rank is defined as the integer rank value of the method at a given iteration among all baselines in terms of regret performance. Both regret and rank are **the lower, the better**. As in existing literature (Müller et al., 2023; Wistuba & Grabocka, 2021), for each benchmark, we report the aggregated regret and aggregated rank, computed as the average regret and rank over five different initializations, with error bars denoting 95% confidence.

All the benchmarks contain a meta-training, meta-validation and meta-test dataset. We train the models with meta-training data. All the hyperparameters of our method (learning rate, acquisition function, MAML inner step size, fine-tuning epochs, etc.) are optimized on the validation set and the results are demonstrated on the test set. PROFBO uses the same pre-trained PFN for all benchmarks. We use an Adam optimizer for PFN training (Kingma & Ba, 2015). Please refer to Appendix B for more detailed experimental settings and comparison.

## 5.2 RESULTS ON REAL-WORLD DRUG DISCOVERY

The docking score, generated through simulation, estimates how strongly a small molecule (ligand) binds to a target receptor (typically a protein) where lower scores indicate stronger predicted

---

[2]Using the same architecture as our PFN, as in Maraval et al. (2023).

[3]OPTFORMER requires datasets with extensive optimization trajectories, which are unavailable in our drug discovery benchmarks.

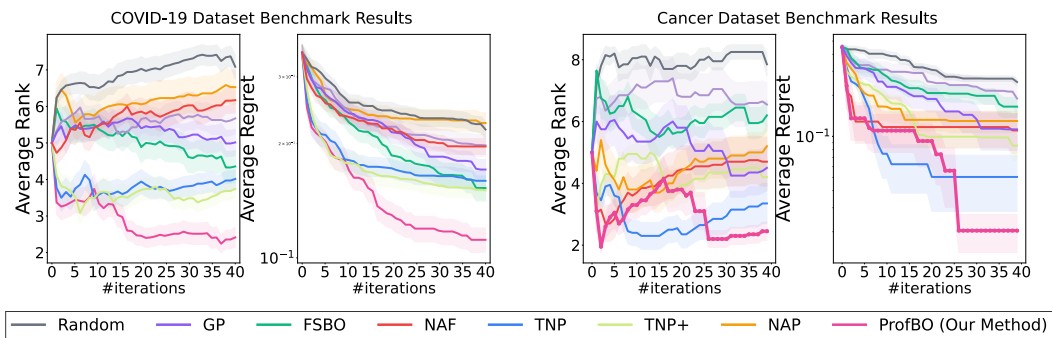

Figure 3: Strong performance of PROFBO on *Covid-B* (five problems) and *Cancer-B* (two problems) compared with other baseline methods.

binding affinity. In drug discovery, a favorable docking score suggests effective interaction with the disease-related receptor, potentially blocking or modulating its function. In this paper, we establish two benchmarks, each consisting of multiple molecule search tasks that aim to minimize the docking score against Covid-19 or cancer receptors, denoted as *Covid-B* and *Cancer-B*. We use a 26D continuous embedding for both datasets, converted from their *mqn feature* (Nguyen et al., 2009). *Covid-B* and *Cancer-B* contain 5 and 2 problems respectively. Check Appendix C for detailed dataset description.

In Figure 3, our method consistently and significantly outperforms baselines and meets the few-shot requirement by achieving strong performance within 40 iterations. In terms of average regret on both datasets, the gaps between PROFBO and other methods are even larger as $T$ increases. While Maraval et al. (2023) reported state-of-the-art NAP 's superiority in a similar antibody design benchmark, in our experiments NAP performs comparably to the lighter MAF, likely due to the challenging high dimensionality of our benchmarks and the Transformer policy's instability in few-shot settings. Notably, NAP required over 200 iterations to dominate in the 11D antibody design task, whereas we evaluate on 26D problems for few iterations.

### 5.3 RESULTS ON HYPERPARAMETER OPTIMIZATION

*HPO-B* (Pineda-Arango et al., 2021) is a benchmark of classification models' hyperparameters and accuracies, widely used in few-shot BO. Following Maraval et al. (2023), we select 6 of the 16 problems, where each corresponds to the loss of a classification algorithm on different tasks, with input dimensions ranging from 2 to 18. We report results over 90 iterations to align with existing literature, though our focus is still on few-evaluation BO. Results on additional 13 problems with 25 iterations are provided in Appendix D, showing the consistent strong performances of PROFBO.

The *HPO-B* results are shown in the left two subfigures of Figure 4. PROFBO achieves strong performances, clearly leading in average rank and regret. Meta-surrogate learning methods (META-GP, FSBO, TNP, TNP+) overlook the procedural knowledge in optimization trajectories, while our surrogate leverages an MDP prior to capture this knowledge, yielding improved results. Among meta-trajectory learning methods, MAF performs even worse than RANDOM, OPTFORMER is competitive, and NAP, also directly modeling trajectories, outperforms both. We attribute this improvement to NAP 's supervised auxiliary loss, which helps the agent better learn inductive biases across tasks. PROFBO balances trajectory modeling and meta-learning by separating the two in a simple yet effective way. MAML enables efficient meta-learning of the trajectory surrogate, while PFN's in-context learning accelerates adaptation to target tasks, yielding stronger performance than NAP and OPTFORMER, especially in the first 10 iterations.

### 5.4 ABLATION STUDIES

To thoroughly understand the algorithmic design of PROFBO, we analyze the effectiveness of three key components in the fine-tuning of PROFBO: the MDP prior, MAML, and positional encoding

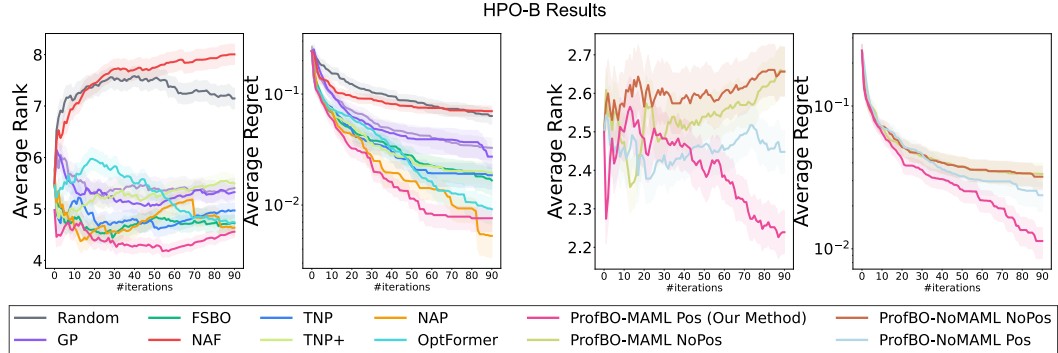

Figure 4: [Left two] Strong performances of PROFBO on *HPO-B* compared with other baseline methods. [Right two] Ablation study results of PROFBO with MAML and positional encoding enabled ("MAML", "Pos") or not ("NoMAML", "NoPos"), showing both MAML and positional encoding are important technical components of PROFBO.

added to PFN. The test setting is the same as Section 5.3. To ensure fair comparison, for each problem, all surrogates are trained with the same dataset pre-generated from our MDP prior.

To test the effectiveness of MDP prior, we compare PROFBO against TNP+. As discussed in Section 5.1, TNP+ is a meta-trained PFN using original meta-data instead of MDP priors, incorporating MAML and positional encoding, so TNP+ serves as the control group. Results shown in Figure 3, 4 provide compelling evidence for MDP priors' great contributions across three benchmarks. Without MDP priors, MAML and positional encoding provide little benefit to Transformer neural processes, as seen with TNP and TNP+[4]. This explains why MAML is effective with MDP priors: they encode richer sequential correlations but are more prone to overfitting, where MAML helps the surrogate extract common optimization patterns. This provides evidence for our claim in Section 4.3.

Next, to study the roles played by MAML and positional encoding, in the right two subfigures of Figure 4, we test four settings where MAML or positional encoding is enabled or not. The figures show that the performances of PROFBO indeed benefit from MAML, as the trials with MAML enabled are better than those without it under the same conditions. Also, vanilla PFN removes the positional encoding to make it permutation-invariant on the context, however, in our task we introduce the positional encoding back to PFN as it can help the surrogate better learn the optimization trajectories, which are also validated in Figure 4.

## 5.5 COMPUTATIONAL EFFICIENCY

NAP (Maraval et al., 2023) is one of the state-of-the-art methods in few-shot BO, and it implements a similar Transformer model to ours. NAP's efficiency was investigated in the original paper, taking only 2% of OPTFORMER's training time. However, PROFBO is even more efficient than NAP.

Both PROFBO and NAP use only the Transformer's forward pass during test, resulting in similar test time, so we only compare the training time of them on the same device implementing a same Transformer architecture. NAP trains the model end-to-end, whereas PROFBO employs a two-stage training paradigm. We evaluate training time across five *Covid-B* problems, each with approximately 1M evaluations. NAP takes 3,925 seconds to finish the process, while PROFBO takes only 1,176 seconds, including 1,045 (MDP prior training) and 131 (fine-tuning) seconds, therefore, PROFBO is *3.34 times faster* than NAP. We attribute PROFBO's efficiency to its lightweight RL agent (a MLP with hidden size 200-200-200-200), which trains faster than NAP's Transformer policy. Additionally, supervised learning in PROFBO is less computationally intensive than RL in NAP.

---

[4]Consistent with Wistuba & Grabocka (2021), which showed that FSBO without MAML often outperforms its MAML-enhanced variant.

## 6 CONCLUSION

In many real-world black-box optimization scenarios, such as drug screening in wet labs, traditional BO methods often fail to converge within a practical number of iterations due to costly evaluations. To solve this problem, we introduce PROFBO, a Bayesian framework that incorporates MDP priors to transfer procedural knowledge from source tasks. Integrated with PFNs and MAML, PROFBO can quickly adapt to the target task in few shots. Experiments on drug discovery and hyperparameter tuning tasks demonstrate consistent improved performances of PROFBO over existing methods. Moreover, its modular design and efficient training process make PROFBO practically ready for a wide range of critical applications. Overall, this work explores the potential of applying principled Bayesian inference to the optimization trajectory prior of real-world experiments, paving the way for more efficient and generalizable optimization strategies that further accelerate scientific discovery.

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

# A  IMPLEMENTATION DETAILS OF PROFBO

## A.1  SETTINGS OF THE PFN TRAINING

The Transformer architecture (Vaswani et al., 2017) processes sequential data by embedding each element into a vector representation and employs a specialized attention mechanism to enable elements to attend to one another during the forward pass. PFNs utilize a tailored attention mask to allow context points in $D$ to attend to each other and be attended by test locations during the forward pass.

The implementation of the PFN is adapted from the repository of PFNs4BO (Müller et al., 2023) [5]. Here we list the settings for the PFN used in our experiment (Table 2). Similar to Müller et al. (2023; 2022), our model takes contextual variables with dimension $\leq 26$ by conducting a zero-padding to the missing dimensions and normalization to the existing dimension. We use a discrete bar distribution for the regression head of PFN, with 1,000 bars uniformly chosen from the interval $[-4.5, 4.5]$.

| Hyperparameters | Choices |
|---|---|
| Embedding size | 512 |
| Number of self-attention layers | 6 |
| Number of heads | 4 |
| Activation function | GeLU |
| Feed forward NN size | 1024 |
| Number of discretized bars | 1000 |
| Pre-training learning rate | {1e-3, 1e-4, 3e-4, 1e-5} |
| Fine-tuning learning rate | 3e-4 |
| Batch size | 128 |
| Sequence length | 40 |
| Steps per epoch | 100 |
| Fine-tuning epoch | 2 |
| Maximum input size | 26 |
| Pre-training prior | GP |
| Acquisition function | {EI, PI, UCB} |
| Optimizer | Adam + Cosine Anneal |

Table 2: PFN settings & Hyperparameters of PROFBO

## A.2  SETTINGS OF THE RL TRAINING

We train our RL agent with the vanilla DQN algorithm, where there is a policy and target Q-network with identical architecture. The target network is a copy of the policy network initially, but it is updated at a specific frequency during the training process. The policy network is used to sample the trajectories, while the target network is treated as the "ground-truth", and is used to calculate the Bellman equation as mentioned in Section 4.

In the main paper, where all baselines are discrete search problems, we developed an adapted training paradigm for the MDP prior to enable efficient learning from extensive molecule meta-data, demonstrating strong empirical performance in generating prior trajectories. Specifically, we restricted the agent to interact with only 10% of the problem, updating to a new 10% subset and updating the target network every 50 epochs. Initially, training multiple RL agents with a dynamic action space per epoch was challenging. In this case, increasing the action space size slowed training without significant gains. Our approach ensures robust increases in cumulative reward while substantially reducing computational costs. Other training hyperparameters of the MDP pirior are shown in Table 3. We note that the feed-forward NN of the agent has the same hidden size as the popular meta-learning neural acquisition methods (Hsieh et al., 2021; Volpp et al., 2020).

---

[5] https://github.com/automl/PFNs4BO

The sampling scheme of the final policy $\pi_\eta^{(i)}$ (i.e., $Q_\eta^{(i)}$) for task $i$ is $\epsilon$-greedy to balance exploration and exploitation:

$$a_t = \begin{cases} \text{Uniform}(\{x_\tau^{(i)}\}_{\tau=1}^{n_i}), \text{Pr} = \epsilon, \\ \arg\max_{a' \in \{x_\tau^{(i)}\}_{\tau=1}^{n_i}}, Q_\eta^{(i)}(s_t, a'), \text{Pr} = 1 - \epsilon. \end{cases}$$

| Hyperparameters | Choices |
|---|---|
| Hidden size | [200, 200, 200, 200] |
| Activation function | ReLU |
| Episode length ($T$) | 40 |
| Epoch | 250 |
| Target update frequency | 50 |
| Discount factor ($\gamma$) | 0.98 |
| Maximum action sample size | $n_i//10$ |
| History size ($k$) | 10 |
| Replay buffer size | 10000 |
| Learning rate | 1e-3 |
| Optimizer | Adam |

Table 3: RL setting of PROFBO

## B  EXPERIMENTAL SETTINGS OF OTHER BASELINES

**FSBO** [6]. FSBO meta-trains a deep kernel GP surrogate with the meta-dataset and performs few epochs of adaptation in each BO iteration. We adopted a deep kernel GP used in Wistuba & Grabocka (2021). We trained the model with batch size 512 for 2000 iterations, so the total number of training data point is 1,024,000, the same as PROFBO. The learning rate is 1e-3.

**NAP** [7]. We use a Transformer with the same architecture as PROFBO (Table 2). The training completely follows the settings mentioned in the original paper, where we perform RL on a conditional GP surrogate trained with meta-data and test on discrete problem. We use a different episode length, epochs, batch size for each benchmark, so that the episode length $T$ matches the experimental results in Section 5 and the total training data is around 1M, please refer to our repository for details.

**OptFormer** [8]. The results in *HPO-B* (Section 5.3) is produced by the authors of NAP based on the official codebase. Please refer to their paper for details.

**TNP.** We used the same PFN architecture with a positional encoding as in Table 2. As mentioned in Section 5.4, we used 1M raw samples from meta-dataset to train the surrogate model with MAML and positional encoding.

**Meta-GP.** Like Maraval et al. (2023), we also pre-train the RBF kernel parameters of the GP with the meta-dataset and initialize the GP with pre-trained parameters at the test time. The implementation is based on BO package BOTORCH (Balandat et al., 2020).

**NAF.** Please refer to Table 3 for the settings of the RL agent. We used a RL prior that learns from all the mata-datasets, instead of a single dataset. To do that, we train the RL agent to optimize $\mathcal{D}^{(i)}, i \in [N]$ for many $i$. Specifically, we sample $i$ uniformly from $[N]$ in each outer loop and train the agent in the same way as described in Appendix A.2.

---

[6] https://github.com/machinelearningnuremberg/FSBO
[7] https://github.com/huawei-noah/HEBO/tree/master/NAP
[8] https://github.com/google-research/optformer

# C EXPERIMENTAL DETAILS OF BENCHMARKS

## C.1 HPO-B

Please refer to official repository [9] and paper Pineda-Arango et al. (2021) for detailed explanation of the dataset. The six representative problems in Section 5.3 are 5860 (glmnet), 4796 (rpart.preproc), 5906 (xgboost), 5889 (ranger), 5859 (rpart), 5527 (svm). We also provide the aggregated results of 13 *HPO-B* problems in the following section of appendix (Figure 8).

## C.2 COVID-B AND CANCER-B

The *Covid-B* dataset comprises five problems as in Pineda-Arango et al. (2021) across 24 tasks. Each task within a problem optimizes the docking score for a shared Covid-19 receptor but targets different binding interactions. For instance, task NPRBD_6VYO_AB_1_F involves interactions with both chains A and B of the PDB structure 6VYO, while NPRBD_6VYO_A_1_F focuses solely on chain A. The validation set consists of unique datasets from Liu et al. (2023) that are distinct from their parent structure. The *Cancer-B* dataset differs from *Covid-B*, as it lacks related tasks from a common parent structure. Instead, it includes docking scores for 5 distinct cancer-target receptors, manually divided into source and target tasks. The validation set is a mixed random sample drawn from each dataset. The final dataset comprises three training sets and two test sets.

For both datasets, the problem for each task is a set of molecules, with size ranging from 2K to 150K and are stored as Simplified Molecular Input Line Entry System (SMILES). We first convert the molecules to their *mqn feature* (42D, integer) (Nguyen et al., 2009), then we use principle component analysis (PCA) to construct a 26D continuous representation for each molecule, where the percentage of explained variance in PCA are greater than 95% for all problems.

**Covid-B.** The *Covid-B* dataset is adapted from the Covid dataset used in DrugImprover (Liu et al., 2023) [10]. They were chosen by the authors from the Zinc 15 dataset (Sterling & Irwin, 2015), containing 11M drug-like molecules. The original dataset consists of 24 .csv files, each containing 1,000,000 molecular docking samples across various SARS-CoV-2 receptors. We identified five structural conditions shared by multiple receptors, each defined as a single *problem* (in *HPO-B*, one learning algorithm corresponds to one problem). While different *datasets* in *HPO-B* share the same supervised loss but differ in training data, in *Covid-B* datasets correspond to the same structural condition under varying sub-conditions. Finally, we retain only the molecules common across their respective problems in each dataset. All the objective values are normalzied to [0, 1].

| Problems | Meta-training | | Meta-test | |
|---|---|---|---|---|
| | # Evaluations | # Datasets | # Evaluations | # Datasets |
| NPRBD | 46,176 | 3 | 30,784 | 2 |
| NSP10-16 | 202,254 | 1 | 202,254 | 1 |
| NSP15 | 11,540 | 5 | 4,616 | 2 |
| Nsp13.helicase | 244,361 | 1 | 244,361 | 1 |
| RDRP | 106,266 | 3 | 35,422 | 1 |

Table 4: Statistics of each *Covid-B* problem in the meta-training and meta-test datasets.

We provide a list of problems of *Covid-B* and the size of each meta-train and meta-test dataset in Table 4. All the problems use a common meta-validation dataset, containing 4 datasets chosen as subsets of 3CLPro_7BQY_A_1_F, NSP16_6W61_A_1_H, PLPro_6W9C_A_2_F, NSP10_6W61_B_1_F and 1M evaluations in total.

**Cancer-B.** We also refer to Liu et al. (2023); Sterling & Irwin (2015) for the original five sets of molecules docked on different cancer proteins. In *Cancer-B*, we utilize three meta-training datasets (6T2W, NSUN2, RTCB), comprising 437,634 evaluations in total, and two meta-test datasets (WHSC, WRN), totaling 291,756 evaluations. The meta-validation set is constructed as a balanced mixture of

---

[9]https://github.com/machinelearningnuremberg/HPO-B
[10]https://github.com/xuefeng-cs/DrugImprover

random samples from these five datasets, totaling to 10,000 evaluations, with each dataset contributing an equal proportion. Molecules are also common in five `.csv` files.

# D  ADDITIONAL EXPERIMENTAL RESULTS

In this section, we present the additional experimental results.

## D.1  ADDITIONAL BASELINE COMPARISON

In the following, we demonstrate the baseline comparison of regret and rank in each problem, i.e., the results in Section 5.2, 5.3 before aggregation w.r.t. problems.

Figure 5 demonstrates the results on 5 problems in *Covid-B*, where PROFBO shows superior performance in general, while NAP is the most unstable one. Figure 6 demonstrates the results on 2 problems in *Cancer-B*, where PROFBO demonstrates robust performances by being the best method after 30 iterations. Other methods suffer from instability in different problems.

Figure 7 shows the results on 6 *HPO-B* problems. We can see that the baselines' performance stabilizes after 90 iterations, and PROFBO is consistently in the top 3 baselines except for Problem No. 5859. This shows that PROFBO is also prominent in problems that require long iterations. In Figure 8, we demonstrate the aggregated results of PROFBO on 13 problems of *HPO-B* for 25 iterations. We find PROFBO still excels in these problems, providing more solid evidence of its ability to adapt to problems with varying input dimensions and diverse meta-task dependencies.

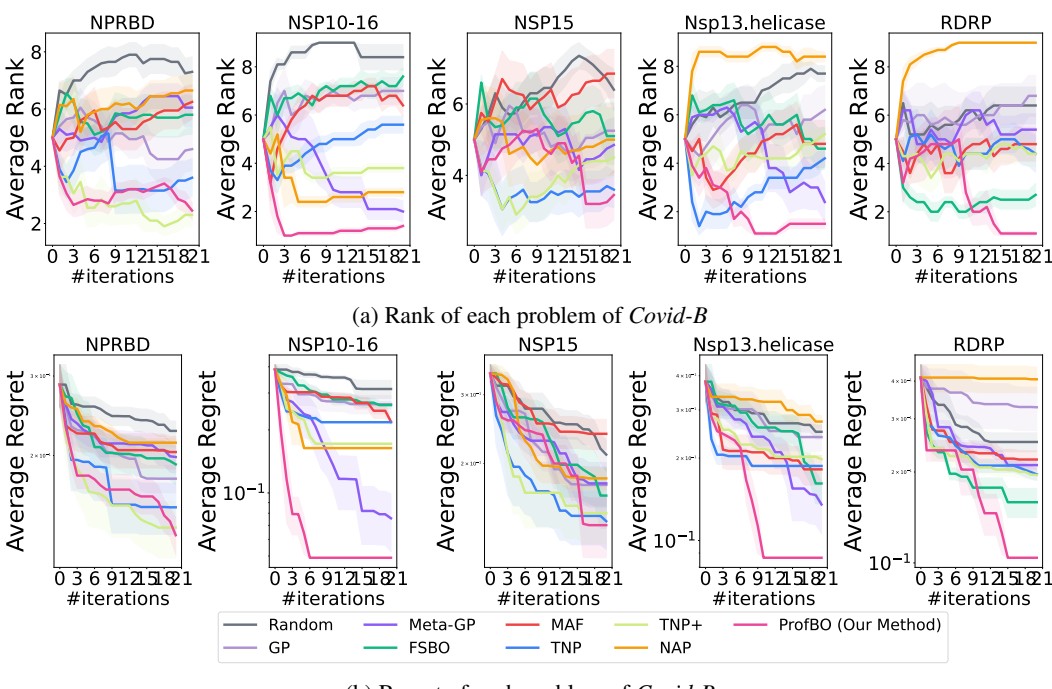

(a) Rank of each problem of *Covid-B*

(b) Regret of each problem of *Covid-B*

Figure 5: Performance comparisons on *Covid-B*.

## D.2  ADDITIONAL FUNCTION PREDICTION COMPARISON

In Figure 9, we demonstrate more visualizable comparison between PROFBO, FSBO, and GP, similar to Figure 1. In addition, we add META-GP to our comparison where the GP parameters are pretrained with meta-datasets. The settings for PROFBO, FSBO and GP are the same as mentioned in Figure 1. As we can see, PROFBO consistently models the true objective function much better than other methods, with only 3 evaluation points.

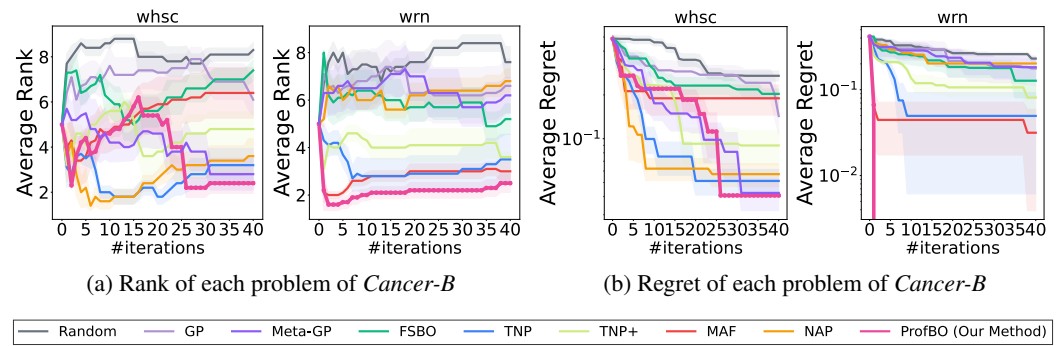

(a) Rank of each problem of *Cancer-B*    (b) Regret of each problem of *Cancer-B*

Figure 6: Performance comparisons on *Cancer-B*.

(a) Rank of each problem of *HPO-B*

(b) Regret of each problem of *HPO-B*

Figure 7: Performance comparisons on *HPO-B*.

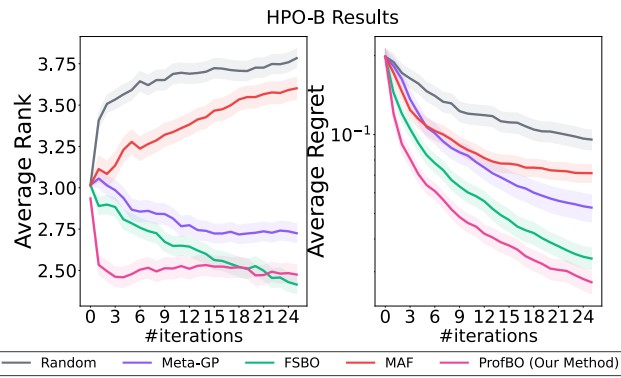

Figure 8: Aggregated results of *HPO-B* with 13 problems for 25 iterations.

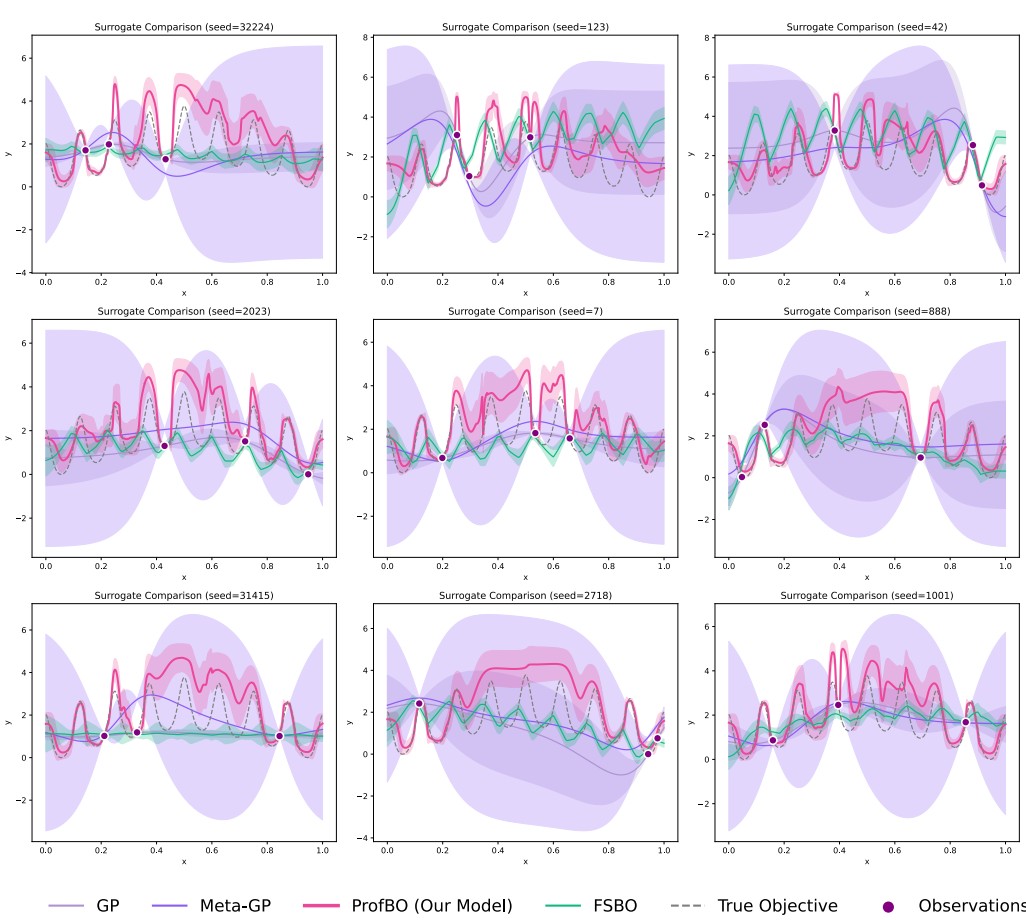

Figure 9: Function prediction comparison.

### D.3 NEGATIVE/UNRELATED TRANSFER

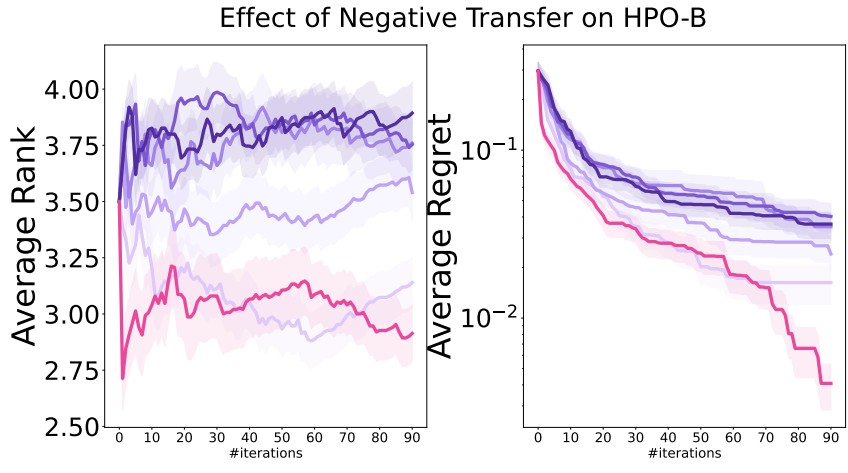

(a) Aggregated results of all problems before & after different levels of negative transfer.

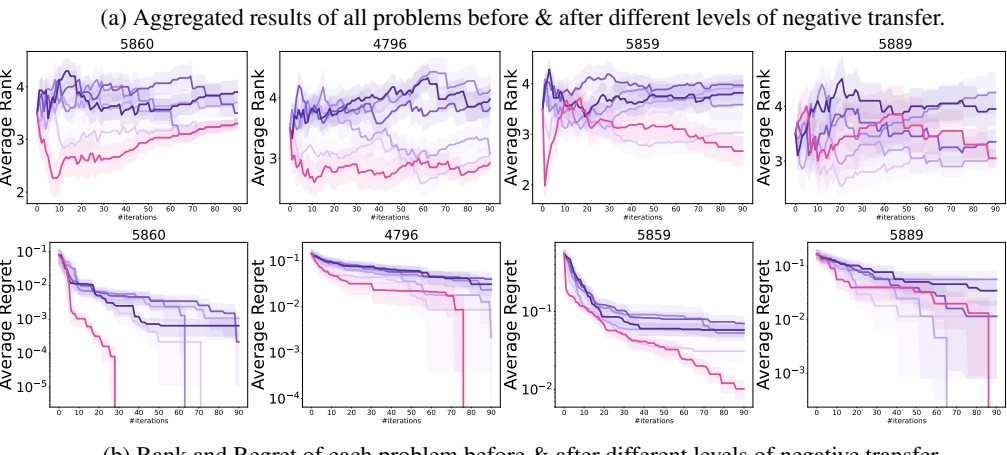

(b) Rank and Regret of each problem before & after different levels of negative transfer.

— negative_epoch_1 — negative_epoch_2 — negative_epoch_3 — negative_epoch_4 — negative_epoch_5 — Our Method (Before Negative Transfer)

Figure 10: A study of the effect of negative transfer on PROFBO tested with problems No. 5860, 4796, 5859, 5889 on *HPO-B*. "negative_epoch_" refers to the negative transfer epochs a surrogate is tuned with. The learning rate of negative transfer is 3e-4. The acquisition function is chosen to be PI. Other training parameters are described in Table 2.

We study how unrelated source tasks will affect the performance of PROFBO. We select 4 totally different hyperparameter optimization tasks with varying input dimensions $d$ from *HPO-B*: problems No. 5860 (glmnet, $d = 6$), 4796 (rpart.preproc, $d = 3$), 5859 (rpart, d= 6), 5889 (ranger, $d = 6$). We tune the surrogates used in Section 5.1, Figure 4 with meta-train data from other tasks. Specifically, we attack the surrogate of problem 5859 with data from 5889, and attack 5889 with 5859, attack 5860 with data of 4796, and attack 4796 with 5860.

We demonstrate the performance of the surrogates on their original tasks after 1 to 5 negative tuning epochs (more epochs indicates stronger negative transfer, Figure 10). On average (Figure 10a), PROFBO's performance is not significantly damaged compared with the surrogate before negative transfer under a moderate negative effect (with a single negative transfer epoch), while it is significantly worse under a stronger negative effect (with more than one negative tune epochs). Moreover, the effect of negative transfer is severer when the discrepancy between unrelated tasks is larger, e.g.,

when the input dimension does not match, which is the case for problems 5860 ($d = 6$) and 4796 ($d = 3$) in Figure 10b.

## D.4 NUMBER/DIVERSITY OF SOURCE TASKS

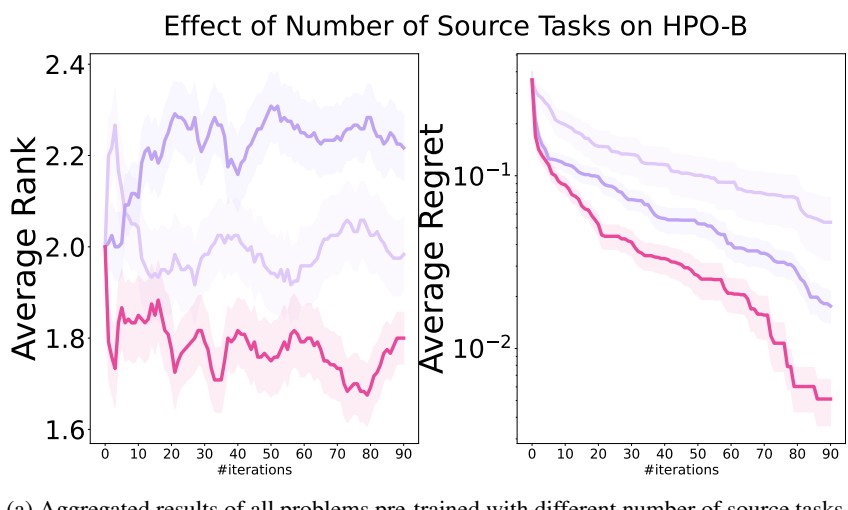

(a) Aggregated results of all problems pre-trained with different number of source tasks.

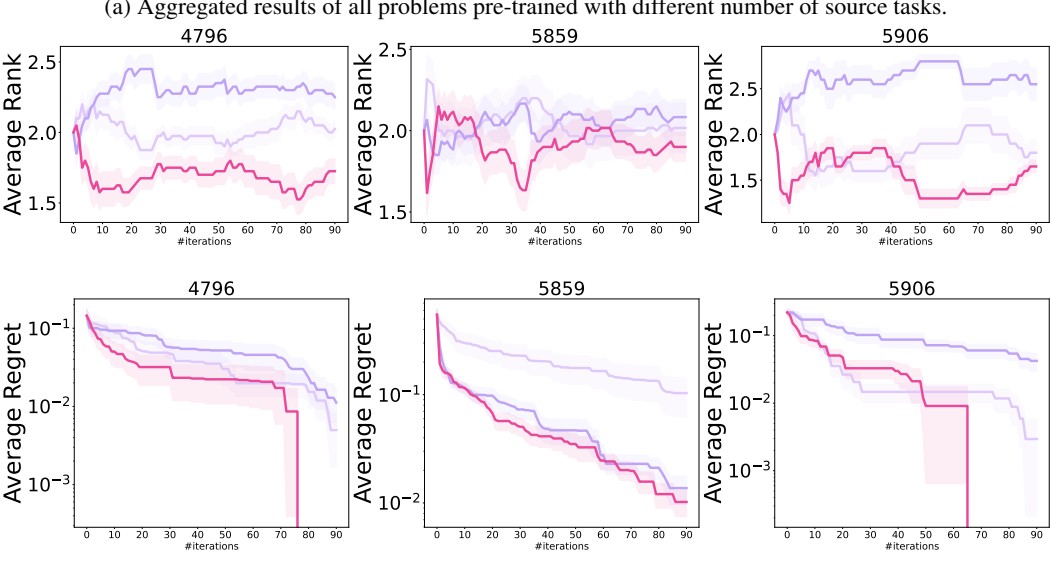

(b) Rank and Regret of each problem pre-trained with different number of source tasks.

Figure 11: A study of the effect of number (diversity) of source tasks on PROFBO tested with *HPO-B* problems No.4796, 5859 and 5906. $10\%, 50\%, 100\%$ are the percentages of the available source tasks used to train the MDP prior and fine-tune the PFN. The tasks of each proportion are independently and randomly sampled. Acquisition function is PI. See Section 5.1 for other evaluation details.

We study the effect of number of sources tasks used to train the MDP prior and fine-tune the PFN on the final performance of PROFBO. We choose problems No. 4796 (rpart.preproc, $d = 3$), 5859 (rpart, $d = 6$), 5906 (xgboost, $d = 16$) from the *HPO-B* benchmark and re-evaluate the performance under $10\%, 50\%, 100\%$ (original one) of available source tasks' data for each problem. The results are shown in Figure 11.

In terms of average effect (Figure 11a), we can observe a clear pattern where increased number and diversity of source tasks indeed correlates with better performance. From Figure 11b, task with higher dimension (5906) seems to be more sensitive to the number of source tasks.

## D.5 GP-LIKE PRE-TRAINING

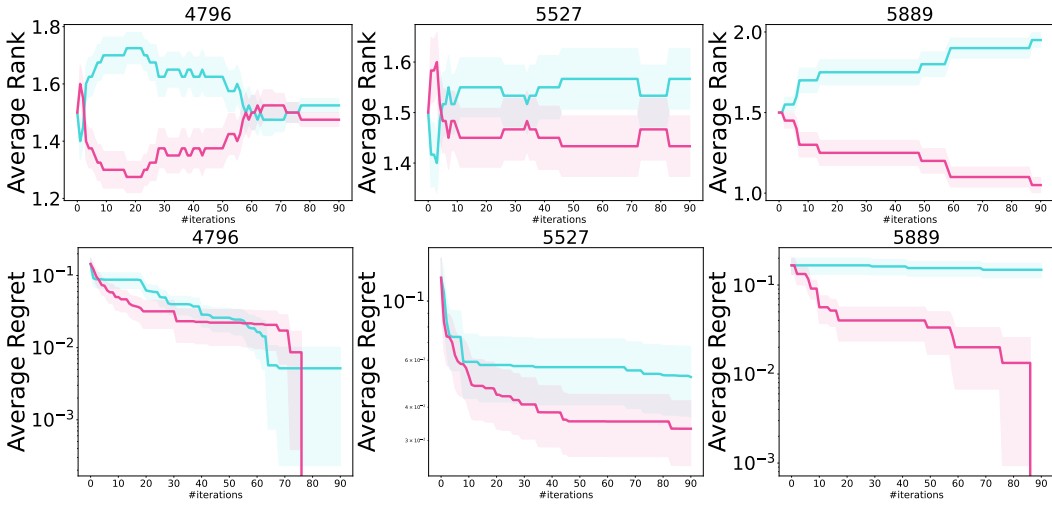

(a) Aggregated results of all problems with or without GP-like Pre-training in Lines 1-4 of Algorithm 1.

(b) Rank and Regret of each problem with or without GP-like Pre-training in Lines 1-4 of Algorithm 1.

Figure 12: A study of the effect of GP-like pre-training (Lines 1-4 of Algorithm 1) on PROFBO performance tested with *HPO-B* problems No. 4796, 5527, 5889. "Only MDP prior" refers to the PFN surrogate trained only with MDP prior data as in Lines 5-12 of Algorithm 1. Acquisition function is PI. See Section 5.1 for evaluation details.

In Section 4, we claimed that the GP-like pre-training described in lines 1-4 of Algorithm 1 will help the PFN make valid inference in the region unexplored by the MDP prior. To test this claim, we now evaluate the performance of PROFBO trained only with MDP prior data, i.e, only conducting Line 5-12 of Algorithm 1 on problems No. 4796 (rpart.preproc, $d = 3$), 5527 (svm, $d = 8$), 5889 (ranger, $d = 6$) from *HPO-B*. The results compared with our proposed full algorithm are shown in Figure 12.

From the aggregated and task-wise results, it is clear that without GP-like pre-training, PROFBO's performance is significantly worse.

## D.6 EXTENDING #ITERATIONS ON *Covid-B* TO 40

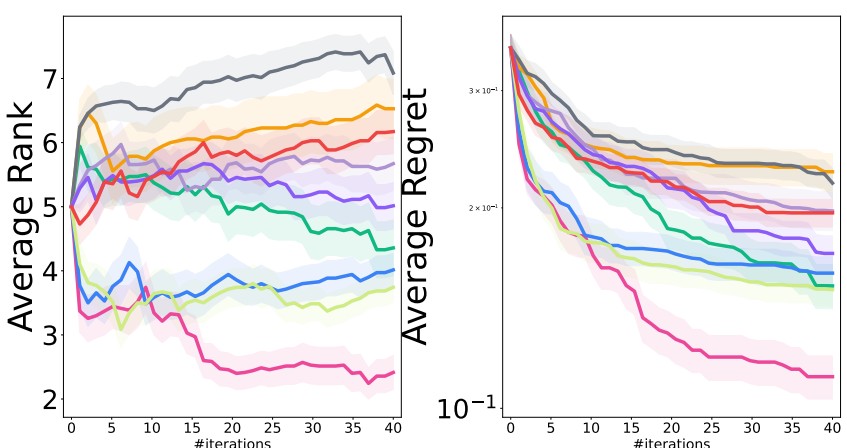

(a) Baseline comparison in *Covid-B* with 40 iterations, aligning with the result in *Cancer-B*.

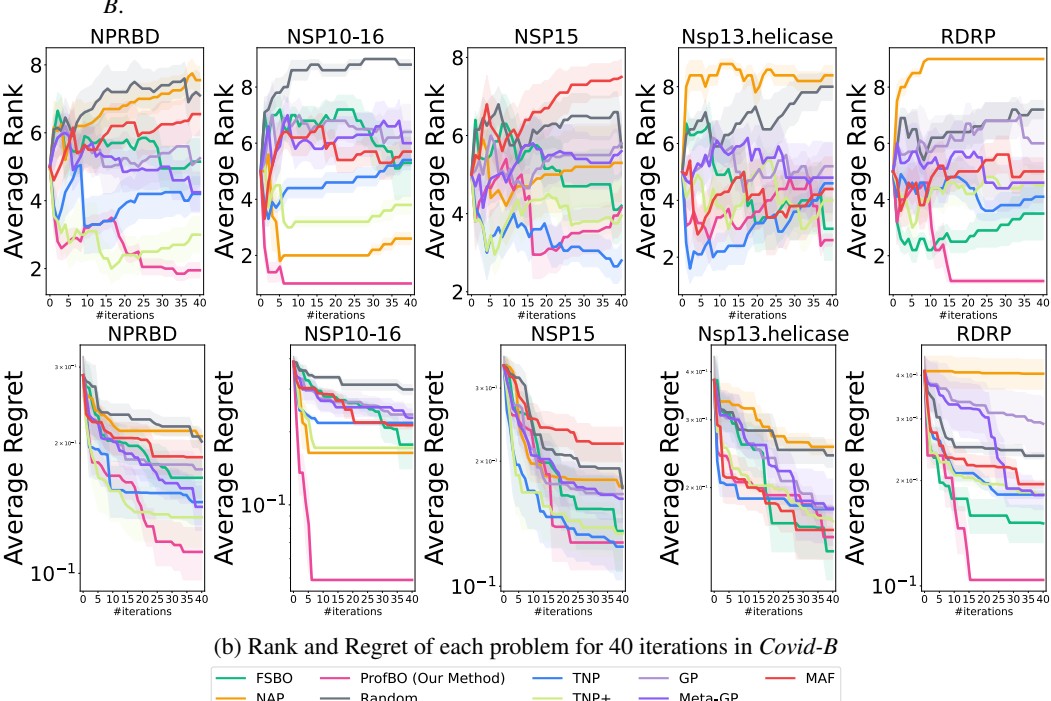

(b) Rank and Regret of each problem for 40 iterations in *Covid-B*

Figure 13: Baseline comparison in *Covid-B*, with #iterations from 0 to 40. This is an extended result from Figure 3 (Right), where only 20 iterations are recorded. See Section 5.1 for evaluation details.

In Figure 13, we extend the results of baseline comparison on *Covid-B* in Section 5.2 from 20 iterations to 40 to better algin with the evaluation on *Cancer-B* (Figure 3, Right).

As shown in Figure 13a, our claim in Section 5.2 about PROFBO's few-shot ($\leq$ 20 iterations) performance on *Covid-B* is still valid on the aggregated result over all problems, and PROFBO still demonstrats great performance within 40 iterations, although there is a certain case (problem NSP15) when TNP outperforms PROFBO after 20 iterations (Figure 13b).

## D.7 ABLATION STUDY FOR MAML AND POSITIONAL ENCODING ON *Covid-B* AND *Cancer-B*

(a) Ablation study for *Covid-B*

(b) Ablation study for *Cancer-B*

Figure 14: Ablation study results of PROFBO with MAML and positional encoding enabled ("MAML", "Pos") or not ("NoMAML", "NoPos"), conducted on *Covid-B* and *Cancer-B* benchmarks, showing both MAML and positional encoding are important technical components of PROFBO. This is an extension of the ablation study described in Section 5.4, which is only conducted on *HPO-B* benchmark (check this section for evaluation details).

As shown in Figure 14, we follow the procedure described in Section 5.4 to conduct ablation study for the contribution of the technical components positional encoding and MAML on *Covid-B* (Figure 14a) and *Cancer-B* (Figure 14b) benchmarks. The result is consistent with the one in Figure 4 (Right), where the surrogates trained with positional encoding generally outperforms those without, and "MAML Pos" gives the significantly best result. This further strengthens our claim about how PROFBO captures positional correlation of optimization trajectories through positional encoding and avoid overfitting to spurious temporal correlation through a MAML-like meta-learning style.

## D.8  TASK SIMILARITY

| Problem | Task Similarity | | | Final Ranking |
|---------|-----|--------|------|------|
| | **Low** | **Medium** | **High** | |
| 5860 (2) | 0.296 | 0.444 | 0.259 | 1 |
| 4796 (3) | 0.222 | 0.444 | 0.333 | 1 |
| 5527 (8) | 0.235 | 0.647 | 0.118 | 2 |
| 5859 (6) | 0.268 | 0.643 | 0.089 | 5 |
| 5889 (6) | 0.350 | 0.400 | 0.250 | 3 |
| 5906 (13) | 0.083 | 0.500 | 0.417 | 2 |

(a) Task similarity of source tasks to the target task of each problem in *HPO-B*. Numbers in "()" are input dimensions.

| Problem | Task Similarity | | | Final Ranking |
|---------|-----|--------|------|------|
| | **Low** | **Medium** | **High** | |
| NPRBD | 0.000 | 0.000 | 1.000 | 1 |
| NSP10-16 | 0.000 | 0.000 | 1.000 | 1 |
| NSP15 | 0.000 | 0.400 | 0.600 | 1 |
| Nsp13.helicase | 0.000 | 0.000 | 1.000 | 1 |
| RDRP | 0.000 | 0.000 | 1.000 | 1 |

(b) Task similarity of source tasks to the target task of each problem in *Covid-B*.

Table 5: Similarity of source tasks to the target task of each problem in *HPO-B* and *Covid-B* and PROFBO's final performance. For both tables, "Problem" is the name of a problem in the benchmark; "Task Similarity" is the proportion of source tasks with Low, Medium, High similarities to the target task of the corresponding problem; "Final Ranking" is the final ranking of regret performance of PROFBO among all baselines (see task-wise results in Figure 7,13b). The calculation of task similarity is based on the misranked pairs count proposed by Falkner et al. (2018).

To study how similarity of source tasks to the target tasks affect the performance of PROFBO, we adopt the notion of misranked pairs count proposed in Falkner et al. (2018), defined for source task $f^{(i)}$:

$$\mathcal{L}(f^{(i)}) = \mathbb{E}_{f^{(i)}}\left[\sum_{q=1}^{n}\sum_{r=1}^{n}\mathbf{1}\{(f^{(i)}(x_q) \leq f^{(i)}(x_r)) \oplus (y_q \leq y_r)\}\right], f^{(i)} \sim \mathcal{GP}^{(i)}$$

where $\bigoplus$ is the "exclusive or" operator; $\mathcal{GP}^{(i)}$ is a GP model fitted with meta-data of $f^{(i)}$; $(x_i, y_i), i = 1, \cdots, n$ are the evaluations of the target task. Thus, $\mathcal{L}(f^{(i)})$ can be seen as the expected frequency of mismatched ranking pairs between the target task value $y_p$ and the prediction of $y_p$ based on source task $f^{(i)}$ evaluated at $x_p$. **The lower the $\mathcal{L}(f^{(i)})$, the more similar the source task is to the target task.** Empirically, we choose $n = 100$, and define "High" similarity in Table 5 as $\mathcal{L}(f^{(i)}) < 3300$, i.e., less than $\frac{1}{3}$ pairs are mismatched. Similarly, we define "Low" similarity as $\mathcal{L}(f^{(i)}) > 6600$ and "Medium" as being between them.

As shown in Table 5, the distribution of similarity of source tasks to the target task varies accross different problems in *Covid-B* and *HPO-B*. Within the results of *HPO-B* (Table 5, (a)), there is a weak correlation between higher task similarity and better final performance. Lack in highly similar task in problem No. 5859 (8.9%) might explain why PROFBO fails to outperform other baselines in this problem. Other factors such as input dimension may also affect PROFBO's performance.

Higher task similarity indeed suggests better final performance of PROFBO **across benchmarks**. As shown in Table 5, (b). *Covid-B* has more highly similar source tasks compared to *HPO-B*, providing intuition of why PROFBO is uniformly the best in *Covid-B*.

## D.9 IMPLEMENTATION ON CONTINUOUS DOMAIN

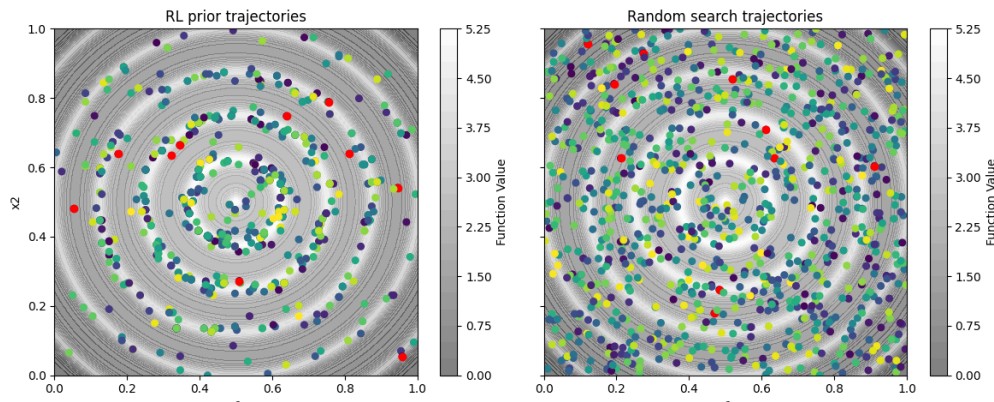

Figure 15: Contour plot of a 2D continuous Ackley function at $[0,1]^2$. $x_1, x_2$ are values of two input dimensions, the funciton value is indicated by the gray shades (the lighter, the higher). [Left] Trajectory samples from a MDP prior. Red points are initializations; Lighter (yellow) points are queried in later iterations. The MDP prior gradually finds optimal solution at $(0,0)$. Moreover, its queries concentrate on local optima. [Right] Random trajectories in the same function.

In Figure 15, we provide a small example of MDP prior's performance on a 2D continuous Ackley function, showing that it is still possible to extract procedural experience of optimization using MDP prior, even for a continuous problem. The training of MDP prior follows the hierachical gridding method used in MetaBO (Volpp et al., 2020). As described in Section 4 and Appendix B.1 of Volpp et al. (2020), a dynamic subset of candidate for the $\xi_t \subset \mathcal{X}$ is chosen at iteration $t$, which is the union of a sparse global discretization $\xi_{global}$ that spans the whole searchspace and a local adaptive subset $\xi_{local,t}$ generated from local discretized searches of first $k$ best evaluations in $\xi_{global}$.

This approach balances between global search of best reward and local exploration using the current RL agent, and has been applied in many meta-learning acquisition methods, including FSAF (Hsieh et al., 2021) and NAP (Maraval et al., 2023). Using this method, Maraval et al. (2023) showed it is possible to train an RL for high-dimensional continuous problems, as they tested NAP on a 135-dimensional mixed integer programming problem. PROFBO can also be applied to continuous problems by this method.

## D.10 ALIGNMENT WITH REAL-WORLD NEED

| Problem | Proportion (20 iter) | Proportion (40 iter) |
|---|---|---|
| NPRBD-6VYO_CD_1_F | 0.8 | 1.0 |
| NPRBD-6VYO_DA_1_F | 0.8 | 1.0 |
| NSP10-16-6W61_AB_2_F | 1.0 | 1.0 |
| NSP15-6W01_A_2_F | 1.0 | 1.0 |
| NSP15-6W01_A_3_H | 0.2 | 0.2 |
| Nsp13.helicase-m3_pocket2 | 1.0 | 1.0 |
| RDRP-7BV1_A_1_F | 1.0 | 1.0 |

Table 6: The proportion of trials in which PROFBO identifies a molecule within the top 0.5% of docking scores for each target task. "Problem" denotes the target task, and "Proportion" indicates how many of the 5 independent trials successfully found a top-0.5% ranked molecule.

In real-world virtual screening workflows, a fixed proportion of top-ranked compounds is typically selected for downstream experimental investigation. For SARS-CoV-2 drug-discovery pipelines

(corresponding to our *Covid-B* benchmark), the **top 0.25%–1%** of compounds is commonly considered promising for follow-up, as reported in Budipramana & Sangande (2022). We evaluate how frequently PROFBO identifies molecules within the **top 0.5%** of docking scores on each Covid-B target task. For every task, we report the proportion of random seeds where PROFBO successfully discovers a top-0.5% molecule within **20** and **40** optimization iterations. The resuts are summarized in Table 6, showing that **PROFBO reliably identifies scientifically meaningful candidates**. On the majority of tasks, the method reaches top-0.5% molecules within 20 iterations, and by 40 iterations achieves 100% success in all but the most challenging case.

These results indicate that 40 iterations on Covid-B is indeed "few-shot optimal," as ProfBO is already able to identify the molecules targeted by human experts within 40 iterations.

## E  STATEMENT OF LIMITATIONS

While ProfBO shows strong empirical performance, our study also reveals several open challenges that extend beyond the scope of this work. First, systematically generating MDP priors remains an unsolved problem. Although we successfully trained MDP prior for *HPO-B*, *Covid-B*, and *Cancer-B* benchmarks using DQN, developing a principled and general methodology that scales across domains, dataset sizes, and mismatched task dimensions is an important direction for future research. Second, although our MAML-based adaptation demonstrates significant performance improvement (Section 5.4), establishing theoretical guarantees for such adaptation mechanisms in meta-BO settings remains an open challenge. Finally, while our experiments clearly indicate that modeling temporal correlations improves sequential optimization (Section 5.4), the underlying nature of these correlations, and how they enhance decision-making remains understood. Addressing these limitations may offer valuable theoretical insights, guide the design of more robust BO algorithms, and clarify when PROFBO is expected to be most effective.

