# OpenReview forum: "None to Optima in Few Shots: Bayesian Optimization with MDP Priors"
_ICLR.cc/2026/Conference — Submitted to ICLR 2026_

### Official Review · Reviewer_NAp6 · 2025-10-22

**Soundness:** 3
**Presentation:** 3
**Contribution:** 3
**Rating:** 6
**Confidence:** 3

**Summary:**

This paper presents ProfBO (Procedure-informed Bayesian Optimization), a framework that accelerates black-box optimization by leveraging priors learned from previous optimization trajectories on related tasks. ProfBO models a prior distribution of past optimization trajectories using a Markov Decision Process (MDP), enabling it to transfer knowledge gained from past optimization runs on related tasks to new tasks. The authors evaluate ProfBO on synthetic and real-world benchmarks, demonstrating that ProfBO consistently outperforms baselines from the literature, achieving faster convergence and lower regret within a very small number of function evaluations. This work shows that leveraging knowledge from past optimization of related tasks can substantially improve sample efficiency in Bayesian optimization.

**Strengths:**

Originality:
ProfBO introduces an original and conceptually elegant idea: learning procedural priors over past optimization trajectories to improve few-shot Bayesian optimization. The method treats prior optimization runs as Markov Decision Processes, enabling it to capture the dynamics of optimization itself. This “procedure-informed” view is a novel way to transfer optimization knowledge and differs from existing approaches.

Quality:
The paper is technically strong and empirically thorough. The methodology is well-motivated. Experiments are extensive, covering a large number of benchmark tasks and comparisons against baselines from the literature. ProfBO consistently achieves faster convergence and lower regret across tasks within small evaluation budgets.

Clarity:
The paper is clearly written and well-structured. The main components of the ProfBO method are clearly motivated and explained. The narrative is easy to follow, and the paper effectively communicates both intuition and implementation details.

Significance:
ProfBO demonstrates that procedural priors can improve sample efficiency of Bayesian optimization. This is significant in real-world domains such as drug design, materials design, etc. where sample efficiency is important.

**Weaknesses:**

While ProfBO demonstrates strong relative performance across benchmarks, including real-world-inspired tasks such the COVID and Cancer benchmarks, the paper does not provide discussion or qualitative analysis of the meaningfulness of the obtained solutions. It remains unclear whether the best-found objective values correspond to scientifically relevant or near-optimal outcomes in these applications, or primarily represent improvements over baselines in a normalized performance sense. Adding brief commentary or case examples illustrating how close these solutions are to practically desirable or known-good results would strengthen the empirical claims and highlight the real-world impact of the method.

**Questions:**

1: Could the authors provide discussion or intuition on why ProfBO achieves larger improvements over baselines on certain tasks but smaller gains on other tasks? Maybe leveraging prior optimization trajectories is more useful for some tasks than others? Do the authors have any intuition for why that is or what types of tasks ProfBO might be most useful for?



2: In experiments and method development, did the authors encounter any scenarios when procedural priors from source tasks are poorly aligned with the target? Is it possible that such a scenario could lead to ProfBO having worse performance than other methods rather than better performance?



3: For the biomedical benchmarks, do the best-found objective values correspond to scientifically meaningful or near-optimal solutions? Adding qualitative insight or examples would help contextualize the results.

---

> ### Author Response · Authors · 2025-11-21
> **Initial Response [1/2]**
>
> Thanks for appreciating the originality, quality, clarity, and significance of our work! We’ve addressed some common concerns in the global response, and we provide point-by-point responses for other concerns below.
>
> > The paper does not provide discussion or qualitative analysis of the meaningfulness of the obtained solutions in real-world benchmarks.
>
> Thank you for the suggestion! We provide a discussion in the following:
>
> 1. Relevance to Drug Discovery Benchmarks
> Our drug-discovery benchmarks reflect the practical task of drug screening, where docking and binding affinity are standard computational proxies for assessing molecular interaction with a target protein. Docking evaluates whether a molecule can adopt a plausible binding pose—**an essential prerequisite for therapeutic activity** [3]. Molecules that cannot form a reasonable configuration are unlikely to modulate a target’s biological function. Our strong few-shot results align with the real-world constraints of screening workflows; see **Appendix D.10 of the revised paper for additional context.**
>
> 2. Impact on Protein Design and ML-Based Drug Delivery
> Our MDP prior offers a principled way to improve sample efficiency within ML-driven protein and molecular design workflows. This is particularly beneficial when experimental validation is costly and slow. For example, even basic ML-guided wet-lab antibody design experiments cost around USD 9,600 per round as reported by [ABTIQUE](https://www.abtique.com/), and obtaining binding measurements for a single protein typically requires 3–4 weeks as reported by [adaptybio](https://start.adaptyvbio.com/). Incorporating our MDP prior can help reduce both cost and turnaround time in such workflows.
>
>
> > Any example to show the obtained results in drug discovery benchmarks align with existing results? What’s its scientific interpretation?
>
> Thank you for raising this important point. We agree that contextualizing the objective values in terms of real-world scientific relevance helps clarify the practical impact of our method.
>
> In real-world virtual screening workflows, _a fixed proportion of top-ranked compounds_ is typically selected for downstream experimental investigation. For SARS-CoV-2 drug-discovery pipelines—corresponding to our Covid-B benchmark—the **top 0.25%–1%** of compounds is commonly considered promising for follow-up, as reported in [here](https://pharmacia.pensoft.net/article/89812/).
>
> To directly address the reviewer’s question, we evaluate how frequently ProfBO identifies molecules within the **top 0.5%** of docking scores on each Covid-B target task. For every task, we report the proportion of random seeds where ProfBO successfully discovers a top-0.5% molecule within **20** and **40** ProfBO optimization iterations:
>
> |problem|proportion of finding top 0.5% at 20 iter|proportion of finding top 0.5% at 40 iter|
> |---|---|---|
> |NPRBD-6VYO_CD_1_F|0.8|1.0|
> |NPRBD-6VYO_DA_1_F|0.8|1.0|
> |NSP10-16-6W61_AB_2_F|1.0|1.0|
> |NSP15-6W01_A_2_F|1.0|1.0|
> |NSP15-6W01_A_3_H|0.2|0.2|
> |Nsp13.helicase-m3_pocket2|1.0|1.0|
> |RDRP-7BV1_A_1_F|1.0|1.0|
>
> These results show that **ProfBO reliably identifies scientifically meaningful candidates**.
>
> These results indicate that 40 iterations is Covid-B are indeed “few-shot,” as ProfBO is already able to identify the molecules preferred by human experts.
>
>
> > Could the authors provide discussion or intuition on why ProfBO achieves larger improvements over baselines on certain tasks but smaller gains on other tasks?
>
> Please refer to Appendix D. 3 - D. 10 in the revised paper for additional experiments, where we discussed factors that influence the ProfBO performance, including quality of source tasks, number of source tasks, and task similarity. In general, ProfBO performs better with higher source task similarity and diversity. Lower input dimension may also help.

---

> ### Author Response · Authors · 2025-11-21
> **Initial Response [2/2]**
>
> > Do the authors have any intuition for why that is or what types of tasks ProfBO might be most useful for?
>
> Thank you for proposing this interesting discussion!
>
> First of all, ProfBO will be useful when there is high-quality and abundant source task data. To implement ProfBO, it is essential to train a MDP prior, thus, the tasks with abundant source task data, e.g., early exploration problems in drug discovery with simulated docking or binding affinity data would allow us to train a better MDP prior.  This is also supported by our additional experiment on diversity of source tasks **(Appendix D. 4)** and task similarity **(Appendix D. 8).**
>
> There is also high potential to implement ProfBO in large-scale problems. As demonstrated in our experimental result (Sec. 5), ProfBO demonstrated superior performance in two drug discovery problems, which have relatively high input size and large dataset size. ProfBO helps capture the procedure information of optimization, essentially reducing the cost of exploring the distribution of the whole response surface to various trajectories. This could be especially beneficial for problems whose search space is large.
>
>
>  >Did the authors encounter any scenarios when procedural priors from source tasks are poorly aligned with the target? Is it possible that such a scenario could lead to ProfBO having worse performance than other methods rather than better performance?
>
> We point to our discussion of task similarity in **Appendix D.8** of the revised paper. With highly mismatched source task data, ProfBO may fail to outperform all baselines in certain problems.
>
> **Reference**
>
> 3. **Frey, N. C., Hö̈tzel, I., Stanton, S. D., Kelly, R., Alberstein, R. G., Makowski, E., ... Gligorijević, V.**
>     _Lab-in-the-loop therapeutic antibody design with deep learning._ bioRxiv, 2025.

---

> > ### Comment · Reviewer_NAp6 · 2025-11-26
> >
> > Thank you for the detailed rebuttal and for answering each of my questions and addressing my concerns. In particular, I think that the added analysis demonstrating the scientific significance of the solutions found for real-world tasks significantly strengthens the paper and sufficiently addresses my primary concern. I continue to think this paper should be accepted.

---

> > > ### Author Response · Authors · 2025-11-26
> > >
> > > We are glad to hear that we have addressed the reviewer’s questions and concerns, and that the additional analysis is found to demonstrate the scientific significance of our solutions for real-world tasks, thereby strengthening the paper! We sincerely thank the reviewer for the continued support and we look forward to the possibility of an updated rating. Please let us know if there are any further questions or clarifications we can provide.

---

### Official Review · Reviewer_ei1T · 2025-10-31

**Soundness:** 1
**Presentation:** 2
**Contribution:** 2
**Rating:** 2
**Confidence:** 3

**Summary:**

The paper targets the important problem of black box optimization where only few evaluations are possible, but optimization trajectories of related tasks are available to speedup the optimization. The paper uses a prior-fitted neural network (PFN) as a model in bayesian optimization. The main contribution is a new method that incorporates optimization trajectories on existing tasks as an MDP prior into the PFN while employing MAML for adapting the trained PFN to the target task.

**Strengths:**

* Tackles an important established problem setting highly relevant to the ICLR community
* Usually PFNs are trained with synthetic data, incorporating actual evaluations is an interesting research direction and a good fit for the tackled problem setting
* Combining MAML and PFN is an interesting methodological contribution
* Follows best practices for reproducibility
* Experimental setup, including baselines, protocol, and used datasets described in sufficient detail.
* Sufficient discussion on related works that use optimization trajectories of related tasks
* Clearly structured and understandable writing

**Weaknesses:**

* The paper does not discuss limitations of their method and experimental design prominently
* The ablation study is not convincing. The results of the ablation study as discussed in the text may not hold in the target range of 20 evaluations. Results are very noisy here and their un-ablated algorithm is not the preferred choice. The ablation study is not performed on the covid-b and cancer-b benchmarks. Why?
* The approach does not take into account categorical parameters. How about integer parameters? Limiting its applicability.
* Unclear how the chosen related tasks affect the approach and evaluation. How many related tasks are necessary? Do some approaches work better for different numbers of related tasks? An explicit evaluation of this would strengthen the paper. What about the case of misleading related tasks? How related are you tasks?
* The authors use different #iterations for each benchmark (20, 40, 90). To keep evaluation here consistent, showing up to e.g., 20 everywhere (which is their target setting) would be better. For example, showing 40 evaluations for the cancer dataset, where your method performs worse than TNP for 20 seems hand picked, what was the reason here? See also my comment on the ablation study.
* How were the final hyperparameter settings of your method chosen? The appendix lists a search space of hyperparameters for you method. Did you tune them? How? Did you also tune the baselines?
* Regret plots for Covid / cancer (Figure 3) only have one y-axis label (10-1), not possible to know the difference in the aproaches at all.
* Unclear how beneficial meta-learning across related tasks compares to other speed-up techniques for BO (e.g., multi-fidelity, cost-aware, or use of expert priors). An empirical evaluation would be best, but a discussion in related works is also missing. This is lacking in prior works, too.

Minor
* Line 180 "It [PFNs] outperforms traditional GP (see Figure 1), " While making an argument for use of PFNs over GPs, you are comparing a meta learned PFN to a standard GP, which is not an apples-to-apples comparison.
* A small discussion of NN-based BO in general would be helpful. In particular, a discussion on PFNs, their inherent meta learning capabilities and their existing applications to BO would be beneficial.
* x-axis of Figure 4 is unlegible
* Paper shows normalized regret. How meaningful are the improvements in absolute terms for the respective benchmarking settings?

**Questions:**

* How many seeds were used?
* You follow Maravaal et. al in selecting 6 of 16 HPO-B problems, how were they selected? How would the results look on all the problems?

---

> ### Author Response · Authors · 2025-11-21
> **Initial Response [1/2]**
>
> Thanks for finding our methodological contribution interesting! We’ve addressed some common concerns in the global response, and we provide point-by-point responses for other concerns below.
>
> We hope our rebuttal could address the misunderstandings, especially some that could be addressed in our original submission, and look forward to the potential of an updated rating. Thank you!
>
> > The paper does not discuss limitations of their method and experimental design prominently
>
> Thank you for pointing this out! We added a Statement of Limitations in Appendix E in the revised paper, please kindly check!
>
> In short: ProfBO performs well empirically, but several challenges remain. A general method for generating scalable MDP priors is still lacking. Theoretical guarantees for MAML-based adaptation in meta-BO settings are also unresolved. Additionally, although temporal correlations clearly help sequential optimization, their underlying role is not yet understood. Addressing these issues could yield deeper insights and guide the development of more robust BO algorithms, while they are open challenges that are beyond the scope of this work.
>
> > In the ablation study results are very noisy and ProfBO does not demonstrate better performance in 20 iterations compared to other settings. Performance of ProfBO on Covid-B and Cancer-B.
>
> We are not sure what "results are noisy" mean. If it was the fluctuation of rank performance in the initial iterations, we note that the final value of the Figure 4 (right) already demonstrated a statistically significant difference between ProfBO and other versions.
>
> For the 20-iteration performance. The primary focus of the ablation study is not to show the few-shot performance of the ProfBO, but to study the effect of each technical component. Our experimental result demonstrated the significant contribution of MAML and positional encoding to the BO performance, which already accomplished the goal of the ablation study.
>
> We have put the ablation study for Covid-B and Cancer-B to **Appendix D.7 of the revised paper**, showing claim about positional encoding and MAML are still valid, please kindly check!
>
> > The approach does not take into account categorical parameters or integer parameters.
>
> Our approach can be easily extended to solve problems with categorical/integer parameters by adapting PFNs to categorical inputs, for example, we can use GP priors with mixed-type kernels for mixed-type input(RBF for continuous, Categorical kernel for categorical search spaces) to pretrain the PFNs, this is similar to NAP [8]. The authors of NAP have also provided an implementation of Meta-learning acquisition BO [15] on mixed input problems, which can be used for training our MDP prior. Those methods, with a well-documented public code base at this repository, can be directly adapted to ProfBO.
>
> > Readability issue of Figure 3 and Figure 4.
>
> Thank you for pointing this out! We updated the figures in the revised paper, please kindly check!
>
> > Unclear how beneficial meta-learning across related tasks compares to other speed-up techniques for BO (e.g., multi-fidelity, cost-aware, or use of expert priors). An empirical evaluation would be best, but a discussion in related works is also missing. This is lacking in prior works, too.
>
> Thank you for pointing this out! We acknowledge that those BO methodologies are not discussed in this paper, and they are also very important aspects of the BO community. However, these settings address problems different from ours. Few-shot (meta) BO aims to use related source-task data to accelerate optimization on new tasks, and the HPO-B, Covid, and Cancer benchmarks lack notions of fidelity or cost, making direct comparison with ProfBO non-trivial, explaining why prior work has not done so. Regarding BO with expert priors (e.g., PFNs4BO [9]), their methods rely on additional meta-knowledge about function optima from domain experts, whereas our setting assumes access only to source-task data.
>
> We would also like to point out that ProfBO itself can be easily adapted to those settings, which may be a direction for future studies.
>
> > Line 180 "It [PFNs] outperforms traditional GP (see Figure 1), " While making an argument for use of PFNs over GPs, you are comparing a meta-learned PFN to a standard GP, which is not an apples-to-apples comparison.
>
> Thank you for giving us the opportunity to clarify this! Figure 1 is not meant to demonstrate the few-shot BO performance of our algorithm, but to visually inform the readers that the vanilla GP surrogate can be improved a lot by designing new knowledge-transfer surrogates, such as Deep-Kenel GP (also included in Figure 1) and our surrogate.

---

> ### Author Response · Authors · 2025-11-21
> **Initial Response [2/2]**
>
> > A small discussion of NN-based BO in general would be helpful. In particular, a discussion on PFNs, their inherent meta learning capabilities and their existing applications to BO would be beneficial.
>
> Thank you for asking this! First, we want to clarify that almost all the compared baselines (FSBO, TNP, MAF, NAP, Optformer) are NN-based. As we discussed in the related work part (Section 2). FSBO uses a deep-kernel GP to meta-learn the response surface of the target task by minimizing the marginal log-likelihood on the training data. TNP is essentially replacing the deep-kernel GP with a transformer neural process, and uses the cross entropy loss aggregated over different source tasks. MAF uses a NN-based RL agent to meta-learn the acquisition policy as described in our appendix A.2, following the approach in MetaBO [15] and FSAF [4].
>
> In terms of PFNs for BO, we actually have discussed it in the last paragraph of Section 2, within the range of Transformer-based meta-BO (Line 135-140).  PFNs was not discussed separately because we mentioned it when discussing the Transformer Neural Process [10], which is almost the same model but was proposed close to but slightly earlier than PFN.
>
> We appreciate your suggestion to highlight the existing applications of PFNs for BO, we have added them to the revised paper (Line 138-139).
>
>
> > Paper shows normalized regret. How meaningful are the improvements in absolute terms for the respective benchmarking settings?
>
> Thank you for pointing this out! We provided an additional experiment in **Appendix D.10 of the revised paper**, please kindly refer to it. We provide a real-world interpretation of the results of ProfBO on Covid-B, where ProfBO consistently discovers molecules in the top 0.5% of docking scores in most target tasks within 20–40 steps, satisfying real-world scientific standards to conduct further screening.
>
> > How were the final hyperparameter settings of your method chosen? Did you tune them? How? Did you also tune the baselines? How many seeds were used? How were HPO-B problems chosen? What’s the result of ProfBO on more HPO-B problems?
>
> **Hyperparameter**: We explicitly explained in **Line 359-364 of the first submission** that:
>
> "*All the benchmarks contain a meta-training, meta-validation and meta-test dataset. We train the models with meta-training data. All the hyperparameters of our method (learning rate, acquisition function, MAML inner step size, fine-tuning epochs, etc.) are optimized on the validation set and the results are demonstrated on the test set. ProfBO uses the same pre-trained PFN for all benchmarks. We use an Adam optimizer for PFN training (Kingma & Ba, 2015). Please refer to Appendix B for more detailed experimental settings and comparison.*"
>
> **Thus, we fine-tuned the Hyperparameters in the validation set for ProfBO, including all the baselines.**
>
>
> **Seed**: We mentioned this in **Line 357 of the first submission**. There are 5 seeds for each sub-task for the Covid, Cancer, and HPO-B benchmarks.  We followed exactly the standard described in the HPO-B ([11],, Section 6), the most widely used benchmark for meta-BO literature, to synthesize the Covid and Cancer benchmark, and to train, validate, test all the baselines on them.
>
>
> **HPO-B problem choice**: We mentioned this in **Line 405-406 of the first submission**: "...each [dataset] corresponds to the loss of a classification algorithm on different tasks, with input dimensions ranging from 2 to 18." The problems are chosen to represent different types of machine learning fine-tuning tasks and a wide range of input dimensions. According to the description of the NAP paper (Section 4.1, first paragraph):
>
> *...we selected six representative search spaces. Nonetheless, we chose the search spaces to represent all underlying classification models in the experiment. We also always picked the ones with the least points to focus on the low data regime performance…*
>
> **Results in more HPO-B problems**: We already put the result on 13 HPO-B problems in appendix D.1, Figure 8. And **we mentioned that in Lines 407-408 of the first submission**.

---

> > ### Author Response · Authors · 2025-11-21
> > **references**
> >
> > 4. **Hsieh, B.-J., Hsieh, P.-C., & Liu, X.**
> >     _Reinforced few-shot acquisition function learning for Bayesian optimization._ NeurIPS 34:7718–7731, 2021.
> >
> >
> > 8. **Maraval, A. M., Zimmer, M., Grosnit, A., & Ammar, H. B.**
> >     _End-to-end meta-Bayesian optimisation with transformer neural processes._ NeurIPS 34, 2023.
> >
> > 9. **Müller, S., Feurer, M., Hollmann, N., & Hutter, F.**
> >     _PFNs4BO: In-context learning for Bayesian optimization._ ICML, PMLR:25444–25470, 2023.
> >
> > 11. **Pineda-Arango, S., Jomaa, H., Wistuba, M., & Grabocka, J.**
> >     _HPO-B: A large-scale reproducible benchmark for black-box HPO based on OpenML._ NeurIPS Datasets and Benchmarks Track, 2021.
> >
> >
> > 15. **Volpp, M., Fröhlich, L. P., Fischer, K., Doerr, A., Falkner, S., Hutter, F., & Daniel, C.**
> >     _Meta-learning acquisition functions for transfer learning in Bayesian optimization._ ICLR, 2020.

---

> ### Author Response · Authors · 2025-11-26
> **Invitation for discussion**
>
> Dear Reviewer ei1T,
>
> Thank you again for acknowledging the novelty of our method and for your thoughtful feedback. We hope that our responses and additional experiments have addressed your questions comprehensively. In particular, regarding your main concerns:
>
> 1. We have added a section to discuss the limitations of our method.
> 2. We have provided additional ablation study on Covid-B and Cancer-B benchmarks, further strengthening our claim about the use of MAML and positional encoding.
> 3. We have offered a detailed discussion about the NN-based BO.
>
> Please let us know if there are any further questions or clarifications we can provide. We deeply appreciate your time and effort in reviewing our work.

---

### Official Review · Reviewer_skmq · 2025-11-01

**Soundness:** 2
**Presentation:** 3
**Contribution:** 2
**Rating:** 2
**Confidence:** 3

**Summary:**

The paper introduces the ProfBO method, which is a prior-fitted network model for transfer learning / metalearning in BO. A key aspect of the method is that for a particular transfer learning problem instance, the PFN is fine-tuned using the source tasks to produce an MDP prior that trains the model on optimization trajectories. DQN in particular is used for modeling optimization trajectories, and a MAML-style approach is used for incorporating them into fine tuning.

The focus of the paper is on settings where a small (20ish) number of evaluations can be made on the target task, and so metalearning from prior tasks is important.The paper shows good empirical performance in this setting on some new test problems and standard benchmarks.

**Strengths:**

* The main contribution of the paper is the MDP prior and the associated incorporation of optimization trajectories into a PFN framework. This is novel and quite interesting. The general strategy also seems like it could be useful outside the transfer learning / metalearning setting that is the focus of the paper.

* The paper also introduces new problems that can be used for evaluating transfer learning methods that are based on real-world problems and seem that they will be useful for future work.

* The ablation study does a good job of showing the importance of various aspects of the optimization trajectory learning and providing insight into how that should be handled.

**Weaknesses:**

* Feasibility of learning optimization trajectory policy on source tasks: The method includes learning a DQN policy network on the source tasks during fine tuning. As I understand it, this requires being able to make new evaluations of the source tasks, and in particular, not just using whatever optimization trajectory you happen to have from some earlier optimization on this task. Is that correct? The typical assumption in metalearning for BO is that the data you have from each source task come from some previous run of BO or the like on that source task, so the number of points per source task would be in the neighborhood of 20-40. How many datapoints per source task were used here? Appendix C.2 says "In Cancer-B, we utilize three meta-training datasets (6T2W, NSUN2, RTCB), comprising 437,634 evaluations in total, and two meta-test datasets (WHSC, WRN), totaling 291,756 evaluations." Does that mean that all O(10^5) points were used as the dataset for learning the DQN on the source tasks? If so, it would present a serious problem for the motivation of the paper. I cannot think of many realistic scenarios where one would be able to run orders of magnitude more evaluations on the source tasks as on the target task. In particular it does not seem to be the case in the scenarios used to motivate the paper: "In practice, these related source tasks can be the docking scores of a set of molecules evaluated on different receptors." If for one receptor (target task) it takes hours/days to do an evaluation, it is implausible that it would take only seconds/minutes to do the evaluation for some different receptor.

* The paper is broadly framed as being for few-shot learning, but it is really about transfer learning. These are not the same thing, generally in few-shot learning one may not have access to the similar source tasks required by the method. Even the title of the paper may be confusing, "None to optima": Except is it really "None" when source tasks are required?

* The potential for negative transfer is an important issue in any transfer learning / metalearning method, and is not investigated in the paper at all. I'd like to see what happens to performance as unrelated tasks are added as source tasks.

* Understanding when trajectory information is helpful: The MDP prior and model for optimization trajectories is the core contribution of the paper. Internally to the paper, the claim that this is important/valuable seems well-supported by the ablation studies, which shows that eliminating either positional encoding or the MAML training algorithm significantly deteriorates performance. But, prior work has come to a different conclusion. The paper for the NAP method (Maravel et al. 2023) has a whole section (Property 3.2) claiming that history-order invariance is important for Meta-RL generally, and that positional encoding should not be used. NAP performs nearly as well ProfBO. So while removing positional encoding degrades ProfBO, other methods (NAP) intentionally exclude positional encoding and yet perform nearly as well as ProfBO with positional encoding. What is the source of this seeming discrepancy?

* The language around optimization trajectories and sampling strategies seems confused. Section 4 states "Under the GP assumption in standard BO, at iteration t, all queries in Dt−1 are treated as i.i.d. uniformly distributed, ignoring the fact that Dt−1 is actually a BO trajectory." This makes it sound like the GP is making distributional assumptions on the X input locations, which is not the case. I think the text is trying to get at the notion of exchangeability, which is a related albeit different concept.

* Number of samples: The paper is framed very strongly around 20 iterations being the upper bound for number of samples. E.g. page 1 "fewer than 20 evaluations"; page 3 "within a few shots, e.g. T<=20"; page 4 "T can be fewer than 20", etc. Then suddenly in Section 5.2 this is changed to "within 20 or 40 iterations." This is of course because the method did not perform particularly well at 20 iterations on the Cancer problem and required 40 iterations to beat TNP. This raises the obvious question of what happens on the covid problem if the number of iterations is taken out to 40, and furthermore seems like there should be some acknowledgement that in some settings more iterations are required.

* It would be helpful to understand when/why more iterations are required. The HPO-B problems were run out to 90 iterations because that's what was done in the work this builds most directly on. We see that there are are significant improvements from iteration 20 to iteration 90. So while it may be correct to say that ProfBO is the best of the methods at iteration 20, it is not correct to say that ProfBO has solved the problem well in fewer than 20 evaluations; the regret is still very high if one were using ProfBO and had to stop after 20 iterations. Is well-performing few-shot learning just not possible here? When is it possible?

**Questions:**

* Please clarify how many points are being used as the dataset for each source task. If this is more than the 20-40 budget used in the target task, please justify.

---

> ### Author Response · Authors · 2025-11-21
> **Initial Response [1/2]**
>
> Thanks for finding our framework design novel and interesting! We’ve addressed some common concerns in the global response.
>
> We believe the reviewer’s main concerns stem from misunderstandings about parts of our paper. In the section “Feasibility of learning optimization trajectory policy on source tasks,” the concerns arise from two misconceptions:
> (1) that training the MDP prior requires new evaluations on the source tasks, and
> (2) a different interpretation of “few-shot BO,” where the reviewer assumes that ProfBO should use only 20–40 evaluations from each source task.
>
> These misunderstandings led to doubts about the real-world feasibility of ProfBO. In our rebuttal, we clarify these points and address each concern in detail. We hope our clarifications help resolve the misunderstandings and lead to a reconsideration of the paper’s evaluation. Thank you!
>
> > Does learning MDP prior from source data require evaluating new objective value from the source tasks?
>
> No, because we can train the RL on already-evaluated source task datasets, which is equivalent to conducting discrete molecule search. Under our problem setting, the evaluated source task data is abundant, providing an environment for the RL agent to learn to optimize source tasks. This also can be seen in the experiment of other RL-based methods such as [5, 8, 15]. In NAP [8], collected optimization  trajectories were used to train the RL agent without querying a new evaluation (Section 4.1 of NAP paper).
>
> > Do all O(10^5) points were used as the dataset for learning the DQN on the source tasks?
>
> No. On average 1/10 data points from the metadatasets are used to train the DQNs. As described in Appendix A.2.
>
> > Is O(10^5) a practical metadataset size for our drug-discovery benchmark?
>
> Yes, this follows the standard in HPO-B [11], the most widely used meta-learning BO benchmark. Each of the HPO-B problem contains $O(10^4 \sim 10^5)$ points (see Table 3 of HPO-B paper), and it is common to conduct meta-training with all the training data available,  while evaluating only $O(10^2)$ (20-40 in our paper) iterations in the test dataset [8-9].
>
> We would also like to clarify: to the best of our knowledge, in meta-BO literatures [2, 4-5, 8, 12, 15-16], **the number of points per source task has never been assumed to be within the number of evaluations on the target task, nor in this paper**. It is only assumed that the target task has limited observation and budget of evaluation, while the source task's observation is abundant.
>
> > The reviewer cannot relate the O(10^5) dataset size to a realistic scenario
>
> The main concern here seems to be the $O(10^5)$ dataset size is not practical in real-world **wet labs**. However, we point out that **Docking scores are generated using simulation**. This is also seen in the Antibody CDRH3-Sequence Optimization benchmark proposed in NAP ([8], Sec 4.2). The simulation-based search for high binding affinity molecules is essential in the exploration stage of drug discovery tasks, as the ability to bind to the target is a prerequisite for all molecules to be therapeutic [3]. It is very typical to compute the docking score with >10^5 molecules.  The initial study using simulated data has already shown to be able to guide the down-stream wet lab experiment, as reported by [ABTIQUE](https://www.abtique.com/):
> .
> “...HyperBind generated 23 high-affinity antibodies (KD ≤ 100 nM) against a challenging GPCR target, including 3 candidates below 10 nM—demonstrating that computational design guided by experimental feedback can achieve binding strengths comparable to therapeutic-grade antibodies.”
>
> Lastly, there are many existing large molecule datasets such as SINC15 [14] (has been applied in [7]) that are ready for pre-training of drug discovery tasks, providing huge potential for us to leverage those information to accelerate the new target tasks.

---

> ### Author Response · Authors · 2025-11-21
> **Initial Response [2/2]**
>
> >  Few-shot BO v.s. Few-shot Learning.
>
> We believe that some of Reviewer’s concerns may stem from a difference in how the term few-shot is interpreted. In our work, following the convention in few-shot BO, few-shot refers to achieving good performance on a target black-box function **within a small number of BO iterations**. In contrast, in traditional few-shot learning, the term often emphasizes adapting to an unseen dataset with very **limited training samples**.
> To avoid confusion, we would like to clarify that in few-shot BO, it is the evaluation budget on the target task that is limited, rather than the sample size of the source tasks. We hope this explanation helps align our intended meaning with the reviewer’s perspective.
>
> We would also like to clarify that the term of “knowledge transfer/meta-learning BO” and “few-shot BO” is mix-used, for example, the few-shot BO literature FSBO [16] is also compared in the literature review of transfer learning for Bayesian optimization [1].
>
> We acknowledge that this could be confusing for the readers. We will change the title of our paper accordingly, e.g., "Meta-learning Bayesian Optimization with MDP priors", addressing the knowledge transfer rather than the "few-shot" learning aspects.
>
> > Discrepancy: Positional encoding is not helpful (NAP [8]) vs. Helpful in this work.
>
> First, we would like to point out that although NAP [8] briefly claimed historical invariance might be useful for BO in Sec. 3.4 of their paper. **The authors did not provide any experimental evidence for this claim, while we provided sufficient ablation study for the usefulness of positional encoding in Section. 5.4 (with further ablation results in Appendix D. 7)**.
>
> Second, as mentioned in Sec. 3.3 of the NAP paper, without positional encoding, they used a uniform random sample for the supervised auxiliary loss, rather than the data collected by RL agent. This aligns with our result in baselines TNP and TNP+ (Sec. 5.4), showing that positional encoding indeed showed no-better performance when the data has no temporal correlation, while we used positional encoding because the MDP prior data has temporal correlation.
>
> Conceptually, it is also not surprising that a sequential decision-making algorithm can benefit from temporal correlations in past observations. This parallels momentum-based optimization, where using previous steps often improves performance. Likewise, leveraging historical “momentum” in our setting may help guide decisions. Although we do not yet have a formal theory for this connection, it suggests a promising direction for future work.
> > Section 4 suggests that GP modeling assumes the query locations are i.i.d. uniform, but GPs make no such assumption on the inputs. The intended point seems to be about exchangeability, which is related but distinct.
>
> Thank you for pointing this out! We acknowledge that the language here could be confusing. We meant to say that under GP assumption, the temporal characteristics of the historical evaluations generated by BO are **ignored** in the next BO iteration. We have corrected this in the revised paper.
>
> >  Claimed 20 iterations is not consistent with experimental results in Sec. 5.2. What’s the performance of ProfBO on Covid-B for 40 iterations?
>
> Thank you for your suggestions! We changed the statements in the paper to a moderate one: within $90$ iterations. We point the experimental results of ProfBO on Covid-B for 40 iterations in the **general response, and Appendix D. 6**.
>
> > ProfBO and other baselines do not find the optimal solution within 20 iterations, how to explain this? Is well-performing few-shot learning just not possible here? When is it possible?
>
> We do not have any general guarantee about when it is possible to sequentially find the optimal solution for a black-box function with noisy evaluation and limited time budget. Even under the assumption of smoothness, there is usually only a high probability asymptotic guarantee [13]. We also note that our claim about ProfBO is to find "high-quality" solution, as in the first paragraph of Sec. 4.
>
> We point out that our approach already demonstrated better performance than other baselines, while none of them is absolutely optimal on average.
>
> We can still observe many instance-level optimal trials of ProfBO and other algorithms in the result (see `HPO-B/results_iclr/Our Method.json` Line 5-15; `HPO-B/results_iclr/NAP.json` Line 1 **in our provided anonymous repository**).

---

> > ### Author Response · Authors · 2025-11-21
> > **References**
> >
> > 1. **Bai, T., Li, Y., Shen, Y., Zhang, X., Zhang, W., & Cui, B.**
> >     _Transfer learning for Bayesian optimization: A survey_, 2023.
> >
> > 2. **Falkner, S., Klein, A., & Hutter, F.**
> >     _Practical transfer learning for Bayesian optimization._ arXiv:1802.02219, 2018.
> >
> > 3. **Frey, N. C., Hö̈tzel, I., Stanton, S. D., Kelly, R., Alberstein, R. G., Makowski, E., ... Gligorijević, V.**
> >     _Lab-in-the-loop therapeutic antibody design with deep learning._ bioRxiv, 2025.
> >
> > 4. **Hsieh, B.-J., Hsieh, P.-C., & Liu, X.**
> >     _Reinforced few-shot acquisition function learning for Bayesian optimization._ NeurIPS 34:7718–7731, 2021.
> >
> > 5. **Iwata, T.**
> >     _End-to-end learning of deep kernel acquisition functions for Bayesian optimization._ arXiv:2111.00639, 2021.
> >
> > 6. **Lee, J., Lee, Y., Kim, J., Kosiorek, A., Choi, S., & Teh, Y. W.**
> >     _Set transformer: A framework for attention-based permutation-invariant neural networks._ ICML 36, PMLR 97:3744–3753, 2019.
> >
> > 7. **Liu, X., Jiang, S., Vasan, A., Brace, A., Gokdemir, O., Brettin, T., ... Stevens, R.**
> >     _DrugImprover: Utilizing reinforcement learning for multi-objective alignment in drug optimization._ NeurIPS 2023 Workshop, 2023.
> >
> > 8. **Maraval, A. M., Zimmer, M., Grosnit, A., & Ammar, H. B.**
> >     _End-to-end meta-Bayesian optimisation with transformer neural processes._ NeurIPS 34, 2023.
> >
> > 9. **Müller, S., Feurer, M., Hollmann, N., & Hutter, F.**
> >     _PFNs4BO: In-context learning for Bayesian optimization._ ICML, PMLR:25444–25470, 2023.
> >
> > 10. **Nguyen, T., & Grover, A.**
> >     _Transformer neural processes: Uncertainty-aware meta learning via sequence modeling._ ICML, PMLR:16723–16738, 2022.
> >
> > 11. **Pineda-Arango, S., Jomaa, H., Wistuba, M., & Grabocka, J.**
> >     _HPO-B: A large-scale reproducible benchmark for black-box HPO based on OpenML._ NeurIPS Datasets and Benchmarks Track, 2021.
> >
> > 12. **Poloczek, M., Wang, J., & Frazier, P. I.**
> >     _Warm starting Bayesian optimization._ Winter Simulation Conference, IEEE, pages 770–781, 2016.
> >
> > 13. **Srinivas, N., Krause, A., Kakade, S. M., & Seeger, M. W.**
> >     _Information-theoretic regret bounds for Gaussian process optimization in the bandit setting._ IEEE Trans. Info. Theory, 58(5):3250–3265, 2012.
> >
> > 14. **Sterling, T., & Irwin, J. J.**
> >     _ZINC 15 – Ligand discovery for everyone._ Journal of Chemical Information and Modeling, 55(11):2324–2337, 2015.
> >
> > 15. **Volpp, M., Fröhlich, L. P., Fischer, K., Doerr, A., Falkner, S., Hutter, F., & Daniel, C.**
> >     _Meta-learning acquisition functions for transfer learning in Bayesian optimization._ ICLR, 2020.
> >
> > 16. **Wistuba, M., & Grabocka, J.**
> >     _Few-shot Bayesian optimization with deep kernel surrogates._ ICLR, 2021.

---

> > > ### Author Response · Authors · 2025-11-26
> > > **Invitation for discussion**
> > >
> > > Dear Reviewer skmq,
> > >
> > > Thank you again for acknowledging the novelty of our method and for your thoughtful feedback. We hope that our responses and additional experiments have addressed your questions comprehensively. In particular, regarding your main concerns:
> > >
> > > 1. We have clarified your concern about the meta-dataset size and provided a solid real-world application scenario for our method.
> > > 2. We have provided additional experimental results for the Covid-19 drug discovery benchmark over 40 iterations, demonstrating consistent performance with our initial claims.
> > > 3. We have conducted additional experiments to evaluate the effect of negative transfer on ProfBO as you requested.
> > > 4. We have offered a detailed discussion of the discrepancy between the use of positional encoding in this study and in NAP, and provided solid experimental evidence supporting our claim.
> > >
> > >
> > > Please let us know if there are any further questions or clarifications we can provide. We deeply appreciate your time and effort in reviewing our work.

---

> > > > ### Comment · Reviewer_skmq · 2025-11-27
> > > >
> > > > I thank the authors for their response.
> > > >
> > > > I still find the motivation of the paper to be a bit unclear. The problem setting of the paper is that we have some specified set of related source tasks with O(10^4) observations, but can only take 20 (or, when required to beat baselines, 40) evaluations on the target task.
> > > >
> > > > The paper gives two examples of real-world problems that the method is meant to handle. They are:
> > > > *  "the docking scores of a set of molecules evaluated on different receptors"
> > > > * "evaluations of the same supervised-learning loss function on different dataset"
> > > >
> > > > In my initial review I raised the first problem, docking scores, and asked how for some receptors we can evaluate 10^4 docking scores, but for the target receptor we can only evaluate ~20. The authors response clarified that these docking scores are actually meant to be used as source tasks when using a different type of evaluation (a wetlab experiment) as the target task.
> > > >
> > > > The "real-world" evaluation in the paper uses docking scores as both source and target tasks, so I hope the authors can understand my missing that point. I certainly appreciate that it is not possible to do an evaluation in the paper with real wetlab experiments, but the extent to which docking scores can actually be used to optimize wetlab evaluations is still a question mark.
> > > >
> > > > In the second problem, tasks are different datasets for training a model. What the authors seem to be proposing is using much smaller datasets as source tasks (so that we can do 10^4 model trainings per dataset) to transfer to a much larger dataset that allows only 20-40 trainings. This makes sense to me in a multi-fidelity BO setting, but there the smaller datasets would normally be downsampled versions of the target dataset. This could be an interesting use for the method in the paper, but the paper is not framed or evaluated as a multi-fidelity BO method.
> > > >
> > > > If not multi-fidelity BO, then I'm not sure why I'd want to use some collection of small datasets for my source tasks instead of downsampling my target dataset. One could suggest that I would do so if the source task optimizations are what I already happen to have lying around, and I don't have evaluations with downsampled target datasets lying around. But if these jobs are cheap enough that I can run 10^4 of them, I can definitely afford to run some downsampled target dataset trainings of the same size, and that just seems like it would be so much more likely to produce similar source tasks and useful transfer.
> > > >
> > > > This is perhaps part of why the literature on meta learning in BO uses source tasks with comparable number of evaluations as the target task. The authors' response claims that this is not the case: "It is only assumed that the target task has limited observation and budget of evaluation, while the source task's observation is abundant." But I continue to disagree on that point. The very first reference provided in the authors' response to support their claim that source task observations are abundant, [2],  says this in Section 6.1: "We optimize each function for a total of 50 iterations. For each meta-task we provide 50 function evaluations obtained by vanilla BO to the transfer HPO method." This is precisely the setting that I described: the source tasks ("meta-tasks" in that paper) have the same (50) number of iterations as the target task, as they are similarly expensive. Similarly, [12] provided by the authors' response states in Section 2 about the source tasks: "we suppose that evaluations are expensive to obtain, limiting the number of evaluations that can be performed to at most a few hundred evaluations per problem". Section 4 of that paper sets the number of observations in the source task equal to the iterations of optimization in the target task (25 for some problems, 50 for another).
> > > >
> > > > That is the classic metalearning-for-BO setting. Now, there is a new family of methods that do pre-training over a large amount of data, and the authors' response references several. But these do not use specifically related source task datasets. For instance [9] does not purport to do metalearning at all, the data for pre-training are all samples from a GP prior. [15] says in Section 5 "We further emphasize that MetaBO does not require the user to manually pick a suitable set of source tasks " The current method seems to operate in an intermediate state: it needs the user to specify a set of related tasks like in classic metalearning approaches, but then expects access to tens of thousands of observations, like with the pre-training methods. I'm just not sure where this exists in the world.
> > > >
> > > > I can certainly imagine scenarios where I have access to cheap simulators of my target problem, which is basically the docking score setting. But that's really a multifidelity BO problem, which is obviously closely related but not identical to metalearning. I think the paper would benefit from using a multifidelity BO framing.

---

> ### Author Response · Authors · 2025-11-28
> **response for reviewer's concern [1/2]**
>
> Thank you for your clarification. We agree that the training setting of Meta-BO, Few-shot BO, meta-learning optimization, etc. varies from early work [2, 12]  to recent work [8, Optformer]. Now, we would like to focus on providing a concrete example that could motivate our drug-discovery benchmarks, and supposedly NAP [8]’s, who also uses a simulation-based anti-body design dataset.
>
> Specifically, the reviewer asks whether there exists a real-world scenario where O(10⁵) source task data is available while only 20–40 target samples are practically obtainable—not as an artificially imposed constraint, but as a genuine limitation.
>
> **Yes, such scenarios exist, and we actually cited in our first response** We highlight the AI-wet-lab loop paradigm in antibody design, exemplified by **Hyperbind** ([ABTIQUE](https://www.abtique.com/) , [HyperBind2: Multi-Shot Learning Enables Progressive Improvement in Computational Antibody Discovery | bioRxiv](https://www.biorxiv.org/content/10.1101/2025.11.06.687005v2)), as a concrete case study.
>
> ---
> Hyperbind operates within an iterative AI-wet-lab loop where computational pre-training leverages massive datasets, while experimental validation is inherently limited to small sample sizes.
>
> **Stage 1: Pre-training on Large-Scale Data (Source Domain)**
>
> As stated in Section 1.2.2 of the Hyperbind paper, large-scale related data was used for pretraining:
>
> _“These models essentially interpolate within dense clouds of existing knowledge, having seen hundreds of similar binding modes and epitope patterns during pre-training.”_
>
> While the exact dataset size is not disclosed, the sources clearly exceed O(10⁵), as we can see from the claimed source of those data(2.3.1):
>
> “
> - Patent Literature Mining: Systematic extraction from global patent databases (2000-2024), capturing marketed therapeutics, optimization trajectories, and negative controls from experimental sections
> - Scientific Literature Curation: Analysis of tens of thousands of publications, particularly
> supplementary materials where negative results are often reported but rarely indexed
> - Proprietary Data Partnerships: Complete experimental campaigns, including the variants that
> typically remain unpublished
> ”
>
> Many publicly available datasets, such as ZINC15 alone contain >10 million compounds.
>
> **Stage 2 : Simulation-Based Screening**
>
> As stated in 2.6:
>
> Using the pre-trained model, Hyperbind performs large-scale virtual screening:
> - Screen **10⁶–10⁸ candidates** from proprietary antibody sequence libraries
> - Rank candidates using structure-aware embeddings learned during foundation training
>
> **Stage 3: Experimental Validation (Target Domain)**
>
> Also in 2.6:
>
> Due to the cost and throughput constraints of wet-lab experiments:
> - Select **~96 diverse candidates** via uncertainty sampling and clustering
> - Validate experimentally using SPR/BLI assays
>
> This O(10¹–10²) sample size for the target task is **not an artificial choice**, it reflects the practical limits of experimental throughput and cost.
>
> ---
>
> ### Clarification on Our Experimental Setup
>
> We use simulated target data because wet-lab experiments are beyond the scope of this computational study. However, in real-world deployment, our framework would operate identically: the target domain would consist of **online experimental measurements**, which are inherently scarce (20–40 samples). A similar experimental setting can be seen in NAP’s anti-body sequence design problem, which also uses a large simulated dataset as source and target data.
>
> ---
>
> ### Real-World Impact
>
> The reviewer questions whether this AI-wet-lab approach has significant practical implications. We note that Hyperbind has been **extensively deployed in commercial applications** by ABTIQUE . As reported on their website:
>
> > *"[Hyperbind] achieves binding strengths comparable to therapeutic-grade antibodies."*
>
> This demonstrates that the AI-wet-lab paradigm: pre-training on large computational datasets followed by few-shot adaptation to experimental data, is not merely a theoretical construct but an **actively deployed industrial approach**.
>
> We sincerely thank the reviewer again for prompting us to provide a solid real world scenario, which could inherently strengthen the motivation of our study.

---

> ### Author Response · Authors · 2025-11-28
> **response for reviewer's concern [2/2]**
>
> **Discussion for Multi-fidelity(MF) settings**
>
> In MFBO settings [*, **], one typically assumes an explicit notion of fidelity across different source tasks, often with an associated cost model. This assumption alone does not apply to our problem. Moreover, MF frameworks usually allow arbitrary querying at different fidelities, while in our setting, as well as in [8, 15], the source data are fixed and there is no predefined fidelity structure.
>
> Methodologically, MF optimization is also fundamentally different from our approach. MF methods rely on the availability of cheap and coarse surrogates of the target task to accelerate convergence. Our method instead introduces an MDP prior and associated knowledge-transfer mechanisms that aim to improve BO performance. The procedural experience we intend to transfer does not align with the assumptions in MF frameworks and cannot be easily adapted to that setting. We have also demonstrated strong empirical performance through our techniques.
>
> Again, the reviewer’s concern about the suitability of our method for the Meta-BO setting mainly relates to the concern about dataset size of the source tasks in our drug discovery benchmark. We have addressed this by providing a concrete example inspired by AI-assisted wet-lab drug discovery workflows, which clarifies the feasibility and motivation of the meta-BO formulation. Even if questions about the drug discovery benchmark persist, the motivation for our Meta-BO problem setting and the way we use source-task data are clearly supported by hyperparameter optimization problems such as HPO-B, which are standard in recent Meta-BO research. From HPO-B [11] (2021)’s benchmark paper, which is after [2] (2018), the training-validation-testing split was explicitly stated in Table 3, which indicates O(10^2-10^4) source task (dataset) evaluation for each source task and in total O(10^3 - 10^5) in total evaluations for each problem (search space).
>
> **reference**
>
> [*] Kandasamy, K., Dasarathy, G., Oliva, J., Schneider, J., & Poczos, B. (2019). Multi-fidelity gaussian process bandit optimisation. Journal of Artificial Intelligence Research, 66, 151-196.
>
> [**] Li, S., Xing, W., Kirby, R., & Zhe, S. (2020). Multi-fidelity Bayesian optimization via deep neural networks. Advances in Neural Information Processing Systems, 33, 8521-8531.

---

### Official Review · Reviewer_AeQn · 2025-11-05

**Soundness:** 3
**Presentation:** 3
**Contribution:** 3
**Rating:** 6
**Confidence:** 3

**Summary:**

The paper proposes PROFBO, a procedure-informed Bayesian optimization framework aimed at solving black-box optimization problems in very few evaluations (typically $T \le 20$). The central idea is to leverage optimization trajectories from related source tasks, modeled as MDP priors, so that the target-task BO does not have to learn an efficient search policy from scratch. Concretely, the method (i) trains lightweight DQN agents on source tasks to generate optimization trajectories, (ii) embeds these trajectory distributions $p(T^{(i)})$ into a prior-fitted Transformer (PFN) to act as a BO surrogate, and (iii) fine-tunes this PFN with MAML and positional encodings to transfer procedural knowledge while avoiding overfitting to specific source-task temporal patterns. At test time the BO loop is standard (context $\to$ posterior $\to$ acquisition), but the surrogate now reflects trajectory-informed priors. Experiments on two new real-world few-shot drug discovery benchmarks (Covid-B, Cancer-B) and on HPO-B show that PROFBO achieves lower regret and better early ranks than meta-surrogate baselines (META-GP, FSBO, TNP) and recent trajectory/meta-BO methods (MAF, NAP, OptFormer), while being more training-efficient than NAP.

**Strengths:**

* Clear and motivated problem setting: BO in regimes where $T \le 20$ and evaluations are expensive, which is where standard asymptotic BO results are less useful.
* Procedural transfer via MDP priors is novel in this particular PFN + MAML setup and allows the surrogate to internalize good search patterns rather than only response surfaces.
* The PFN backbone gives a principled way to do single-pass Bayesian inference over contexts, leading to faster inference than GP posteriors and enabling the use of non-GP priors.
* Ablation studies isolate the contribution of MDP priors, MAML, and positional encodings and show that each of them improves early-iteration regret.
* Strong empirical results on newly proposed real-world discrete/continuous drug-like tasks (Covid-B, Cancer-B) and on HPO-B, outperforming several state-of-the-art meta/few-shot BO baselines.
* Training-time comparison with NAP shows the proposed modular two-stage training (RL for priors, supervised PFN fine-tuning) is more efficient than end-to-end RL with transformers.

**Weaknesses:**

* The approach depends critically on the availability and quality of source-task optimization trajectories; the paper does not quantify how performance degrades when source tasks are few, noisy, or mismatched with the target.
* The MDP prior is learned with per-task DQN on (possibly) large discrete action spaces, and although the authors optimize it (subset of actions, batched GPU generation), this can still be expensive in domains without precollected meta-data.
* The method is benchmarked mostly on structured or tabular/meta datasets with fixed embeddings (26D molecule embeddings, HPO-B); it is less clear how the approach would behave on high-dimensional continuous design spaces where actions cannot be discretized so easily.
* No acquisition-function–level comparison is made under exactly matched hardware/time budgets; some of the reported gains could be due to PFN’s fast forward pass rather than the MDP prior per se.
* There is no formal analysis of negative transfer: in principle a trajectory prior that encodes a suboptimal policy could bias the PFN and hurt few-shot performance.

**Questions:**

1. How sensitive is PROFBO to the number and diversity of source tasks? For example, if only a small subset of Covid-B problems is available for training, does the method still outperform TNP/META-GP on the held-out problems?
2. In Section 4.2, the MDP defines the action space as candidate points from the dataset. How would PROFBO be instantiated for continuous domains where we cannot enumerate actions and cannot easily train DQN on a discrete subset?
3. The PFN head outputs a discretized bar distribution over $y$. When computing EI/UCB from this surrogate, are the baselines given access to the same acquisition budget (number of candidate points scored per iteration)?
4. The two-stage training (GP-like pretrain, then MDP-prior fine-tune with MAML) is argued to prevent overfitting to trajectory order. Can the authors show a small example where training only on trajectory priors actually harms OOD target tasks?
5. For the drug-discovery tasks, can the authors confirm that target tasks are not leaked into the MDP-prior training stage (i.e., that there is a strict meta-train/meta-test split at the trajectory level)?

---

> ### Author Response · Authors · 2025-11-21
> **Initial Response**
>
> We appreciate the reviewer for liking our paper and finding our procedural transfer novel! We’ve addressed some common concerns in the global response, and we provide point-by-point responses for other concerns below.
>
> > Performance of ProfBO when source tasks data are few or unrelated.
>
> Please kindly refer to the results of negative transfer, number (diversity) of source tasks in our **global response**. They can also be seen in **Appendix D.3, D. 4 of the revised paper**.
>
>
> > Training MDP can be expensive in domains without precollected meta-data.
>
> Our experiments have already shown that our training time is lower than that of transformer-RL methods. Also, assuming no precollected meta-data is beyond the scope of meta-BO methods as meta-data is essential for **all** Meta-BO methods to perform well in unseen black-box objectives [2, 4-5, 8, 12, 15-16]. And the goal of this paper is to provide an accelerated BO method under the assumption of abundant meta-datasets (as we stated in Sec. 3.1.).
>
> Moreover, ProfBO is robust under moderate reduction of meta-data. We demonstrate in **Appendix D.4 of our revised paper** that ProfBO's performance is not significantly harmed with half of the training data. This shows ProfBO can still be applied to early experimental trials with less observation available.
>
> > Performance of ProfBO in high-dimensional continuous search space.
>
> High dimensional black-box optimization is usually considered as a hard problem, but our method is not primarily designed to solve this. Instead, our focus is to accelerate BO on an unknown target with source task information. However, it is not hard to adapt ProfBO to continuous domain, we demonstrate through a concrete example in **Appendix D. 9 of the revised paper** that the MDP prior is still applicable for continuous problems.
>
> > No acquisition-function–level comparison is made under exactly matched hardware/time budgets; some of the reported gains could be due to PFN’s fast forward pass rather than the MDP prior per se.
>
> Thank you for pointing this out! For acquisition-level comparison, we would like to clarify that PFN generates a discretized approximation of the posterior distribution, and the acquisitions are computed using the logits, and optimized by choosing a maximum in a finite set, which means there is no significant budget differences in the computation and maximization of acquisition functions.
>
> We are not sure what 'reported gains' means, and kindly request the reviewer to clarify. If the reviewer meant to say the computational efficiency results in Sec 5.5, we point out that NAP and ProfBO both use a Transformer Neural Process model as the base model with the same model structure (as specified in Table 2). Despite trained in supervised learning (ProfBO) or RL (NAP) manner, the type of computation involved in both models during training are almost the same (Transformer forward + backward, as seen in Algorithm 1 of our paper, and Algorithm 1 of NAP), i.e., NAP also enjoys the fast forward pass. We also note that the major training budget of ProfBO comes from the cost of RL (1,045 of 1,176 seconds), and the training of NAP is also a RL procedure, on the same GPU device (Line 464). Thus, it is reasonable to observe a lower time budget of the RL training with a lightweight agent (200-200-200-200) for MDP prior compared to the RL training with a Transformer.
>
> > Are the baselines given access to the same acquisition budget (number of candidate points scored per iteration)?
>
> Yes, we confirm that all the baseline are given access to the same acquisition budget, which is the set of all available candidates. This is ensured by the public code implementation of [HPO-B](https://github.com/machinelearningnuremberg/HPO-B) benchmarks. For drug discovery benchmarks, in each iteration, the acquisition budget is $\min (10000, \text{dataset size})$ for all baselines.
>
> > Is GP-like pre-training (in Line 1-4 of Algorithm 1) useful?
>
> It indeed benefits the algorithm, which can be shown from our new experimental results in **Appendix D.5 of the revised paper**, where we compare the performance of ProfBO trained only with MDP prior with the full version.
>
> > For the drug-discovery tasks, can the authors confirm that target tasks are not leaked into the MDP-prior training stage?
>
> Yes, we confirm that there is no leakage of target tasks into the MDP-prior training stage. All drug-discovery experiments enforce a strict meta-train / meta-test split at the _trajectory_ level, ensuring that no information from the target tasks is used when training the MDP prior.

---

> ### Author Response · Authors · 2025-11-21
> **references**
>
> 1. **Bai, T., Li, Y., Shen, Y., Zhang, X., Zhang, W., & Cui, B.**
>     _Transfer learning for Bayesian optimization: A survey_, 2023.
>
> 2. **Falkner, S., Klein, A., & Hutter, F.**
>     _Practical transfer learning for Bayesian optimization._ arXiv:1802.02219, 2018.
>
> 3. **Frey, N. C., Hö̈tzel, I., Stanton, S. D., Kelly, R., Alberstein, R. G., Makowski, E., ... Gligorijević, V.**
>     _Lab-in-the-loop therapeutic antibody design with deep learning._ bioRxiv, 2025.
>
> 4. **Hsieh, B.-J., Hsieh, P.-C., & Liu, X.**
>     _Reinforced few-shot acquisition function learning for Bayesian optimization._ NeurIPS 34:7718–7731, 2021.
>
> 5. **Iwata, T.**
>     _End-to-end learning of deep kernel acquisition functions for Bayesian optimization._ arXiv:2111.00639, 2021.
>
> 6. **Lee, J., Lee, Y., Kim, J., Kosiorek, A., Choi, S., & Teh, Y. W.**
>     _Set transformer: A framework for attention-based permutation-invariant neural networks._ ICML 36, PMLR 97:3744–3753, 2019.
>
> 7. **Liu, X., Jiang, S., Vasan, A., Brace, A., Gokdemir, O., Brettin, T., ... Stevens, R.**
>     _DrugImprover: Utilizing reinforcement learning for multi-objective alignment in drug optimization._ NeurIPS 2023 Workshop, 2023.
>
> 8. **Maraval, A. M., Zimmer, M., Grosnit, A., & Ammar, H. B.**
>     _End-to-end meta-Bayesian optimisation with transformer neural processes._ NeurIPS 34, 2023.
>
> 9. **Müller, S., Feurer, M., Hollmann, N., & Hutter, F.**
>     _PFNs4BO: In-context learning for Bayesian optimization._ ICML, PMLR:25444–25470, 2023.
>
> 10. **Nguyen, T., & Grover, A.**
>     _Transformer neural processes: Uncertainty-aware meta learning via sequence modeling._ ICML, PMLR:16723–16738, 2022.
>
> 11. **Pineda-Arango, S., Jomaa, H., Wistuba, M., & Grabocka, J.**
>     _HPO-B: A large-scale reproducible benchmark for black-box HPO based on OpenML._ NeurIPS Datasets and Benchmarks Track, 2021.
>
> 12. **Poloczek, M., Wang, J., & Frazier, P. I.**
>     _Warm starting Bayesian optimization._ Winter Simulation Conference, IEEE, pages 770–781, 2016.
>
> 13. **Srinivas, N., Krause, A., Kakade, S. M., & Seeger, M. W.**
>     _Information-theoretic regret bounds for Gaussian process optimization in the bandit setting._ IEEE Trans. Info. Theory, 58(5):3250–3265, 2012.
>
> 14. **Sterling, T., & Irwin, J. J.**
>     _ZINC 15 – Ligand discovery for everyone._ Journal of Chemical Information and Modeling, 55(11):2324–2337, 2015.
>
> 15. **Volpp, M., Fröhlich, L. P., Fischer, K., Doerr, A., Falkner, S., Hutter, F., & Daniel, C.**
>     _Meta-learning acquisition functions for transfer learning in Bayesian optimization._ ICLR, 2020.
>
> 16. **Wistuba, M., & Grabocka, J.**
>     _Few-shot Bayesian optimization with deep kernel surrogates._ ICLR, 2021.

---

> ### Comment · Reviewer_AeQn · 2025-11-26
>
> Thank you for the thorough response and for addressing each of my questions and concerns. I have increased my score :)

---

> > ### Author Response · Authors · 2025-11-26
> >
> > We are glad to hear that we have addressed the reviewer's questions and concerns, and sincerely thank the reviewer for the strong support! Please let us know if you have any further questions.

---

### Author Response · Authors · 2025-11-21
**Global Response**

We highly appreciate **all reviewers** for finding our ProfBO framework **novel and interesting**! We also appreciate the thorough and constructive feedback provided. All suggestions have been incorporated and highlighted in blue in the revised manuscript. In addition, we have included all requested experiments. A summary of these new experiments is provided below, and they consistently reinforce the claims made in our original submission.

| **Experiment** | **Requested by** | **Location** | **Claim** |
|----------------|------------------|--------------|----------------|
| Negative / Non-related Transfer — Fine-tune HPO-B tasks with varying amounts of unrelated meta-data (1–5 epochs). | All reviewers | Appendix D.3 | ProfBO is robust to small amounts of irrelevant meta-data; performance remains stable. |
| Effect of Amount of Source Data — Experiments using 10%, 50%, and 100% of source data. | AeQn, ei1T | Appendix D.4 | For COVID benchmark, using 50% data yields similar performance. |
| Training with MDP prior only, without GP-pretraining. | AeQn | Appendix D.5 | Performance degrades, confirming the need for two-stage training. |
| Extend iterations on Covid-B from 20 to 40. | skmq, ei1T | Appendix D.6 | All baselines converge; ProfBO remains statistically superior. |
| Test MAML and positional encoding separately on Covid and Cancer benchmarks (extended ablation study). | ei1T | Appendix D.7 | Both MAML and positional encoding are needed as parts of ProfBO. |
| Task Similarity Analysis — Compute and compare task similarity (RGPE-style metrics) across several HPO-B problems. | ei1T | Appendix D.8 | Tasks show measurable similarity structure, supporting transfer effectiveness. |
| ProfBO in Continuous Domains | AeQn | Appendix D.9 | ProfBO can be extended to continuous domains, maintaining coherent behavior. |
| Real-world interpretation for drug discovery | NAp6, ei1T | Appendix D.10 | ProfBO consistently discovers molecules in the top 0.5% of docking scores in most target tasks within 20–40 steps. |


To address reviewers’ concerns, we first address some common comments. Point by point responses are given with respect to each review.

> 1. Effect of negative transfer: what if some source tasks are unrelated to the target?

**In Appendix D. 3 of the revised paper**, we study how unrelated source tasks affect the performance of using four heterogeneous HPO-B problems with varying input dimensions. Each surrogate model (as in Section 5.1, Figure 4) is “attacked” by tuning it with meta-train data from an unrelated task. For example, using data from task 5889 to tune the surrogate for 5859, or using data from 4796 to tune the surrogate for 5860. We then evaluate each surrogate on its original task after one to five negative-tuning epochs, where additional epochs represent stronger negative transfer.
On average, ProfBO remains mostly unaffected under mild negative transfer (a single tuning epoch), but its performance degrades when the negative influence is stronger. The severity of negative transfer increases as the discrepancy between tasks grows, and it is particularly pronounced when input dimensions differ, as demonstrated by the interaction between tasks 5860 (6D) and 4796 (3D).

> 2. Effect of number (diversity) of source task on the performance of ProfBO.

**In Appendix D.4 of the revised paper**, we study how the number of source tasks used to train the MDP prior and fine-tune the PFN affects the final performance of PROFBO. Using problems 4796 (rpart.preproc, (d=3)), 5859 (rpart, (d=6)), and 5906 (xgboost, (d=16)) from HPO-B, we re-evaluate performance when only 10%, 50%, or 100% of the available source-task data is used. The corresponding results are presented in Figure 11.
Overall, the average trend in Figure 11a shows that increasing the number and diversity of source tasks leads to improved performance. Figure 11b further indicates that higher-dimensional tasks, such as problem 5906, are more sensitive to reductions in the amount of source-task data.

> 3. The performance of ProfBO on Covid-B extended to 40 iterations.

As shown in **Figure 13a of Appendix D.6 of the revised paper**, our claim in Section 5.2 about ProfBO’s few-shot (≤ 20 iterations) performance on Covid-B is still valid on the aggregated result over all problems, and ProfBO still demonstrates great performance within 40 iterations, although there is a certain case (problem NSP15) when TNP slightly outperforms ProfBO after 20 iterations (Figure 13b).

---

### Meta-Review · Area_Chair_7rB4 · 2026-01-03

**Summary:**

This work proposes a new Bayesian optimization approach called ProfBO. It leverages optimization trajectories of source tasks to improve the performance on target tasks. The key idea is to embed MDP priors into a Prior-Fitted Neural Network and use Model-Agnostic Meta-Learning to adapt to target target tasks.

The reviewers raised a range of concerns, including negative transfer, impact of related source tasks, assumptions on large numbers of points per source task, clarity, lack of discussions on limitations and results, etc. The authors provided detailed rebuttal for those concerns and updated the paper. However, there are a few concerns that I believe were not fully addressed. A major one is whether it is reasonable to assume a very large amount of data exists for source tasks, but only <100 evaluations are allowed on the target task (questions from Reviewer skmq). Other ones include the impact of the chosen related source tasks since the authors did not provide results for evaluating "how related should the source tasks be" and other approaches' performance with fewer related tasks (questions from Reviewer ei1T).

The authors also made the claim that “the number of points per source task has never been assumed to be within the number of evaluations on the target task”, which unfortunately contradicts the papers cited by the authors, as pointed out by Reviewer skmq. I'd suggest the authors add an experiment where the numbers of datapoints from the source tasks are around 100, to match the size of the evaluations on the target task to better explain how well the proposed method works when such assumptions on the source tasks are not made.

Similarly, for the impact of the chosen related source tasks, it'd be good to include other methods' results and a measure of "relatedness", together with how different choices of related tasks impact the performance.

Given that some core concerns still remain unresolved, I recommend rejection.

In the future versions of this work, I also hope to see what the source tasks look like in Figure 1. It is interesting that FSBO did not capture the distribution, given that it learns the amplitude parameter in the squared-exponential kernel (see page 7 of https://arxiv.org/pdf/2101.07667 "Few-Shot Bayesian Optimization with Deep Kernel Surrogates"). It would be nice if the authors would further specify how they set up FSBO since what I can find is only " a deep kernel GP used in Wistuba & Grabocka (2021)" in Section B. Moreover, regarding the necessary number of datapoints in source tasks, there exist results showing that a very small number of datapoints are needed from source tasks in HPO-B (see page 44 of https://jmlr.org/papers/volume25/23-0269/23-0269.pdf "Pre-trained Gaussian Processes for Bayesian Optimization"). It could also be interesting to see whether "Pre-trained Gaussian Processes" may capture the distribution better in Figure 1 or in other problems studied in the paper.

**Reviewer Concerns:**

Reviewer AeQn: The concerns include dependency on the number of source tasks, the fact that MDP prior is expensive to learn, discrete search space and negative transfer. These concerns were addressed.

Reviewer skmq: The key outstanding concern is the assumption that a lot of datapoints, e.g., O(10^5), are available in source task. The authors have shown that they used simulations to get a large number of datapoints in the source task, but this brings a new problem: why not use multi-fidelity approaches to better model the problem.

Reviewer ei1T: The key outstanding concerns are categorical parameters (the authors claim it is easy without showing clear evidence) and impact of the chosen related tasks (explained above). Other ones like discussions on limitations and unconvincing ablation study are probably addressed.

Reviewer NAp6 had concerns about discussion or qualitative analysis of the meaningfulness of the obtained solutions which were addressed by the authors.

**Reviewer Scores:**

Reviewer AeQn already updated score.

Reviewer skmq had discussions with the authors, did not update the score due to remaining concerns.

Reviewer ei1T probably would not have changed the score given the outstanding concerns.

Reviewer NAp6 had discussions and remained positive.

---

### Decision · Program_Chairs · 2026-01-26

Reject